# Task-Specific Data Selection for Instruction Tuning via Monosemantic Neuronal Activations

**Da Ma**$^{\alpha}$, **Gonghu Shang**$^{\alpha}$, **Zhi Chen**$^{\sigma}$, **Libo Qin**$^{\delta}$, **Yijie Luo**$^{\alpha}$,
**Hongshen Xu**$^{\alpha}$, **Lei Pan**$^{\gamma}$, **Shuai Fan**$^{\gamma\mu}$, **Kai Yu**$^{\alpha\mu\lambda}$, **Lu Chen**$^{\alpha\beta\mu\lambda}$*
$^{\alpha}$X-LANCE Lab, MoE Key Lab of Artificial Intelligence, AI Institute
School of Computer Science, Shanghai Jiao Tong University, Shanghai, China
$^{\beta}$Shanghai Innovation Institution, Shanghai, China
$^{\gamma}$AISpeech Co., Ltd., Suzhou, China $^{\sigma}$ByteDance
$^{\delta}$School of Computer Science and Engineering, Central South University
$^{\mu}$Jiangsu Key Lab of Language Computing, Suzhou, China
$^{\lambda}$Suzhou Laboratory, Suzhou, China
{mada123, chenlusz}@sjtu.edu.cn

## Abstract

Instruction tuning improves the ability of large language models (LLMs) to follow diverse human instructions, but achieving strong performance on specific target tasks remains challenging. A critical bottleneck is selecting the most relevant data to maximize task-specific performance. Existing data selection approaches include unstable influence-based methods and more stable distribution alignment methods, the latter of which critically rely on the underlying sample representation. In practice, most distribution alignment methods, from shallow features (e.g., BM25) to neural embeddings (e.g., BGE, LLM2Vec), may fail to capture how the model internally processes samples. To bridge this gap, we adopt a model-centric strategy in which each sample is represented by its neuronal activation pattern in the model, directly reflecting internal computation. However, directly using raw neuron activations leads to spurious similarity between unrelated samples due to neuron polysemanticity, where a single neuron may respond to multiple, unrelated concepts. To address this, we employ sparse autoencoders to disentangle polysemantic activations into sparse, monosemantic representations, and introduce a dedicated similarity metric for this space to better identify task-relevant data. Comprehensive experiments across multiple instruction datasets, models, tasks, and selection ratios show that our approach consistently outperforms existing data selection baselines in both stability and task-specific performance[2].

## 1 Introduction

Instruction tuning [1, 2] enables large language models (LLMs) to better follow human instructions, powering versatile applications such as chatbots [3–6]. However, real-world tasks often require specialized abilities—like professional physics problems—that general instruction tuning may not provide [7]. While fine-tuning LLMs on diverse and broad instruction datasets [8, 9] improves their overall generalization, performance on specific tasks often remains suboptimal [7]. This raises a key challenge: *how can we efficiently select data subsets from general instruction datasets to maximize LLM performance on targeted tasks, given only a handful of examples* (§ 2.1) [10, 11].

---

*Corresponding author
[2]Available on https://github.com/OpenDFM/MoNA

39th Conference on Neural Information Processing Systems (NeurIPS 2025).

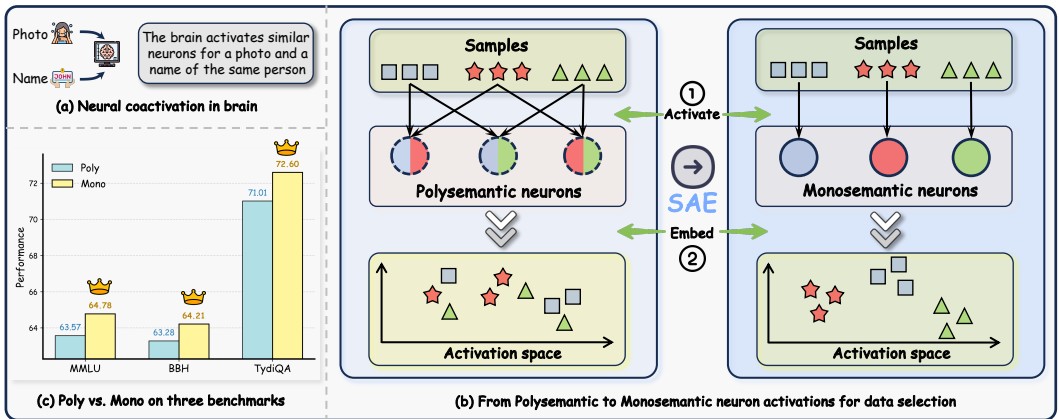

Figure 1: (a) An example of neural coactivation in brain [25] (b) Disentangling polysemantic activations into monosemantic representations via a sparse autoencoder (SAE) (c) Improved data selection using monosemantic activations (More details are in Appendix B.3)

To address this challenge, data selection methods can be roughly grouped into two main categories: *influence-based* methods and *distribution alignment* methods. Influence-based methods [11–13] select training data by estimating the influence of each candidate sample on the evaluation loss for target examples [14], and prioritize samples that are expected to most reduce this loss. However, these approaches can be *unstable* since evaluation loss can fail to reflect true model performance after instruction tuning, causing inconsistency in selection results [15–17]. As an alternative, distribution alignment methods [18–21] select data whose distribution is similar to that of the target task, in order to minimize distribution shift and improve generalization [10, 22–24]. These methods typically embed samples into a feature space, define a similarity metric within this space, and select data most similar to representative samples of the target task (§ 2.2).

While distribution alignment offers a principled framework for data selection, its effectiveness depends heavily on how samples are represented. Existing methods typically rely on *data-centric* embeddings, derived either from shallow textual features (e.g., BM25 [18] and DSIR [19]) or high-dimensional neural representations (e.g., BGE [20] and LLM2Vec [21]) extracted directly from the data. While effective in many settings, such representations may fail to capture the internal computational dynamics of how the model processes different data points, which is crucial for task-specific performance [26, 27]. To address this, we introduce a *model-centric* paradigm which explicitly captures these internal dynamics by *representing each data point through the neuronal activation pattern it triggers in a pretrained model*. This approach is inspired by neuroscience, where related concepts trigger coordinated neural responses [28] (Figure 1-a). Accordingly, we select those training samples whose activation signatures in the model are most similar to those elicited by target task exemplars.

Nonetheless, directly using raw neuron activations can yield suboptimal similarity estimates due to neuron polysemanticity, where a single neuron may respond to multiple, unrelated concepts [3] [26]. As a consequence, unrelated samples can appear spuriously similar due to shared activation of polysemantic neurons (left of Figure 1-b). To mitigate this issue, we employ sparse autoencoders (SAEs) [27, 29, 30] to disentangle polysemantic activations into sparse, monosemantic units, making similarity in activation space more aligned with semantic similarity (right of Figure 1-b, § 2.3). This improvement is illustrated in Figure 1-c, where data selection based on monasemantic activations yields superior performance compared to raw polysemantic activations. In addition, we introduce a dedicated similarity metric tailored to the sparse monosemantic activation space produced by SAEs, as detailed in § 2.4.

In summary, our contributions are: 1) We propose MoNA (Monosemantic Neuronal Activation-based Data Selection), a novel method for task-specific data selection in instruction tuning. MoNA represents each sample using sparse, monosemantic neuronal activations derived from sparse autoencoders,

---

[3]For example, one neuron in a large language model may be activated by both academic citations and HTTP request patterns.

enabling model-centric data selection. 2) We introduce a similarity metric tailored to this sparse monosemantic activation space, enabling more accurate identification of task-relevant examples. 3) Comprehensive experiments across multiple candidate instruction tuning datasets, evaluation tasks, models, and data selection ratios demonstrate that MONA consistently outperforms existing data selection approaches. We will release our code to facilitate further research in the community.

## 2 Methods

We begin this section by formalizing the task-specific data selection problem and its objective. Next, we outline the distribution alignment framework that underpins our approach. We then describe our proposed method (MONA) in detail, including the construction of monosemantic neuronal activation embeddings and the dedicated similarity metric designed for this embedding space.

### 2.1 Problem Formulation

Given a large-scale general instruction dataset $\mathcal{D}^{\text{src}} = \{\mathbf{s}_i\}_{i=1}^{N}$ and a small set of representative examples from the target task $\mathcal{D}^{\text{tgt}} = \{\mathbf{s}_j\}_{j=1}^{M}$, where $M \ll N$, our goal is to select a subset $\mathcal{D}^{\text{sel}} \subset \mathcal{D}^{\text{src}}$ such that fine-tuning a large language model (LLM) on $\mathcal{D}^{\text{sel}}$ leads to the best overall performance on the target task, denoted as $\mathcal{T}^{\text{tgt}}$.

Formally, let $\mathcal{M}_\theta$ denote an LLM with parameters $\theta$, and $S(\mathcal{M}_\theta, \mathcal{T}^{\text{tgt}})$ denote its performance metric (e.g., accuracy, F1-score) on the target task. The data selection objective is formulated as:

$$\mathcal{D}^{\text{sel}} = \underset{\mathcal{D} \subset \mathcal{D}^{\text{src}}, \, |\mathcal{D}|=k}{\arg\max} \; S\left(\mathcal{M}_{\theta^*}, \mathcal{T}^{\text{tgt}}\right), \tag{1}$$

where $|\mathcal{D}| = k$ is a budget constraint on the number of selected samples, and $\theta^*$ is obtained by fine-tuning the model on $\mathcal{D}$:

$$\theta^* = Optimize\left(\mathcal{M}_\theta, \mathcal{D}\right). \tag{2}$$

However, directly optimizing Eq. (1) is computationally infeasible, as it involves enumerating all possible subsets and retraining the model for each. This motivates the need for efficient data selection criteria that can identify high-quality subsets with minimal supervision and computation.

### 2.2 Distribution Alignment Pipeline

A common approach to the data selection problem defined in § 2.1 is *distribution alignment*, which seeks to select a subset $\mathcal{D}^{\text{sel}}$ whose distribution closely matches that of the target examples $\mathcal{D}^{\text{tgt}}$. The objective in Eq. (1) then becomes:

$$\mathcal{D}^{\text{sel}} = \underset{\mathcal{D} \subset \mathcal{D}^{\text{src}}, \, |\mathcal{D}|=k}{\arg\max} \; Sim\left(\mathcal{D}, \mathcal{D}^{\text{tgt}}\right), \tag{3}$$

where $Sim(\cdot, \cdot)$ measures the distribution similarity between the selected and target sets. In practice, this similarity is operationalized by first computing, for each sample in $\mathcal{D}$, its similarity to the target set $\mathcal{D}^{\text{tgt}}$ in the embedding space, and then aggregating these sample-level similarities. Based on this, the pipeline consists of three steps (see the left panel of Figure 2):

1. *Embedding*: Each sample $\mathbf{s}_i \in \mathcal{D}^{\text{src}} \cup \mathcal{D}^{\text{tgt}}$ is projected into a $d$-dimensional feature space via an embedding function $\Phi : \mathbf{s} \mapsto \mathbf{z} \in \mathbb{R}^d$, where $\mathbf{z}_i$ denotes the embedding of $\mathbf{s}_i$.[4]

2. *Similarity Metric Definition*: Explicitly define a similarity metric, denoted as $\delta(\mathbf{s}_i, \mathcal{D}^{\text{tgt}})$, to quantify the similarity between a source sample $\mathbf{s}_i \in \mathcal{D}^{\text{src}}$ and the target set $\mathcal{D}^{\text{tgt}}$.

3. *Subset Selection*: The final subset $\mathcal{D}^{\text{sel}}$ consists of the $k$ samples from $\mathcal{D}^{\text{src}}$ with the largest aggregate similarity to the target set:

$$\mathcal{D}^{\text{sel}} = \underset{\mathcal{D} \subset \mathcal{D}^{\text{src}}, \, |\mathcal{D}|=k}{\arg\max} \sum_{\mathbf{s}_i \in \mathcal{D}} \delta(\mathbf{s}_i, \mathcal{D}^{\text{tgt}}). \tag{4}$$

---

[4]In this paper, $\mathbf{z}_i$ refers to the embedding of sample $\mathbf{s}_i$.

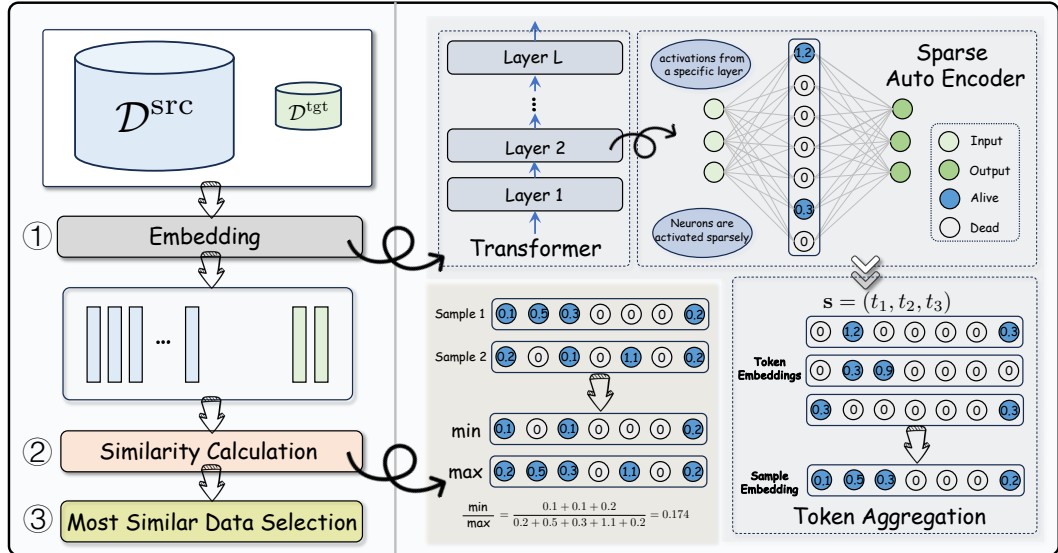

Figure 2: Workflow of MONA. **Left**: Distribution alignment pipeline between the source dataset and the target task. **Right**: Computation of monosemantic neuronal activation embeddings and the proposed similarity metric. *Top*: Application of SAE; *Bottom right*: Aggregation of token embeddings into a sentence-level embedding; *Bottom left*: Calculation of similarity between two samples

## 2.3 Monosemantic Neuronal Activation Embedding

**Intuition** In neuroscience, related stimuli trigger coordinated neuron activations, reflecting semantic similarity [28]. Inspired by this, we hypothesize that neural network activations can similarly capture semantic relationships between samples. To this end, we represent each sample by its activation pattern from *a predefined set of neurons* (e.g., a specific layer)[5] in a neural network $\mathcal{M}^{\text{ds}}$:

$$\mathbf{z} = f_{\text{NAS}}\left(\mathcal{M}^{\text{ds}}, \mathbf{s}\right),\tag{5}$$

where $f_{\text{NAS}}(\cdot, \cdot)$ extracts neuronal activations for input $\mathbf{s}$ from $\mathcal{M}^{\text{ds}}$.

**Sparse Autoencoder-based Monosemantic Decomposition** Building on the intuition, we use the neuronal activation states from a single, predefined layer in the transformer [31] as the basis for the embedding.[6] However, even the activations from a single layer can exhibit polysemanticity, where a single neuron responds to multiple, often unrelated, concepts [26]. For example, one neuron in a large language model may be activated by both academic citations and HTTP request patterns, making such activations difficult to interpret and less effective for representing a specific semantic property [26]. This polysemanticity undermines the interpretability and reliability of feature representations based directly on raw activations.

To address this, we employ a sparse autoencoder (SAE) following [27, 29, 30] (see the top right part of Figure 2). The SAE transforms the original neuron activations into a higher-dimensional, sparse activation space, where each resulting neuron tends to respond to a distinct, monosemantic feature or concept, rather than simply producing a learned representation. Prior work [29] has demonstrated that such sparse activations exhibit improved interpretability and semantic purity, which is beneficial for our data selection framework.[7] Formally, given an input sequence $\mathbf{s} = (t_1, t_2, \ldots, t_n)$ of $n$ tokens, let $\mathbf{h}_k^\ell \in \mathbb{R}^d$ denote the output at layer $\ell$ of model $\mathcal{M}^{\text{ds}}$ for token $t_k$ $(1 \le k \le n)$. The SAE computes a new sparse activation for each token as follows:

$$\mathbf{z}_k^\ell = TopK\left(\mathbf{W}_{\text{enc}}(\mathbf{h}_k^\ell - \mathbf{b}_{\text{pre}})\right),\tag{6}$$

---

[5]Using all neurons in the network would result in an extremely high-dimensional embedding, equal to the total number of model parameters, and bring significant computational overhead.

[6]We compare the effect of different layer choices in § 3.5.

[7]We provide further evidence for this through the visual validation shown in Figure 4.

where $\mathbf{W}_{\text{enc}} \in \mathbb{R}^{d' \times d}$ ($d' \gg d$) and $\mathbf{b}_{\text{pre}} \in \mathbb{R}^d$ are trainable parameters. The operator $TopK(\cdot)$ retains only the largest $K$ values of the input vector and sets all remaining entries to zero.[8] This transformation produces a sparse and interpretable activation pattern for each token in the sequence, enabling more reliable and semantically meaningful representations for downstream data selection.

**Token Aggregation for Sample Embedding**   After obtaining the sparse monosemantic activation $\mathbf{z}_k^\ell$ for each token in the input, we aggregate these token-level vectors by averaging over all tokens to form a sample-level embedding (see the bottom right of Figure 2):

$$\mathbf{z} = \frac{1}{n} \sum_{k=1}^{n} \mathbf{z}_k^\ell, \tag{7}$$

where $n$ is the number of tokens in the input $\mathbf{s}$. Averaging, rather than summing, is crucial for mitigating length bias: without normalization, the selection process systematically prefers samples that match the average length of samples in $\mathcal{D}^{\text{tgt}}$, rather than those with the highest semantic relevance. This bias often harms downstream performance, as demonstrated in Appendix D.5.

In summary, $f_{\text{NAS}}(\cdot, \cdot)$ in Eq. (5) corresponds to the composition of the token-level sparse mapping in Eq. (6) and the aggregation operation in Eq. (7).

## 2.4   Similarity Metric for Monosemantic Neuronal Activation Embedding

This section describes how we define the similarity metric within the monosemantic neuronal activation embedding space for use in the distribution alignment pipeline. The procedure consists of two steps: (i) aggregating the embeddings of the target examples to form a task prototype, and (ii) computing the generalized Jaccard similarity [32] between each source sample and the task prototype.

**Task prototype representation**   To improve efficiency, we aggregate the monosemantic neuronal activation embeddings of all target examples in $\mathcal{D}^{\text{tgt}}$ into a single task prototype. This reduces the computational complexity from $\mathcal{O}(|\mathcal{D}^{\text{tgt}}| \cdot |\mathcal{D}^{\text{src}}|)$ to $\mathcal{O}(|\mathcal{D}^{\text{src}}|)$. Formally, the task prototype is defined as:

$$\mathbf{z}^{\text{tgt}} = \frac{1}{|\mathcal{D}^{\text{tgt}}|} \sum_{\mathbf{s}_j \in \mathcal{D}^{\text{tgt}}}^{|\mathcal{D}^{\text{tgt}}|} \mathbf{z}_j, \tag{8}$$

where $\mathbf{z}_j$ is the embedding of the $j$-th target example.

**Generalized Jaccard Similarity**   For high-dimensional, sparse feature representations such as our monosemantic neuronal activation embeddings, classic similarity metrics such as Euclidean or Cosine similarity can become unreliable due to the "curse of dimensionality" [33]. Our empirical results confirm that neither Euclidean nor Cosine similarity is suitable for this embedding space (see § 3.5 for details). As an alternative, we adopt the generalized Jaccard similarity (see the bottom left part of Figure 2). Mathematically, given a source sample $\mathbf{s}_i$ with embedding $\mathbf{z}_i$ and the task prototype $\mathbf{z}^{\text{tgt}}$, the generalized Jaccard similarity is defined as:

$$\delta\left(\mathbf{s}_i, \mathcal{D}^{\text{tgt}}\right) = \frac{\sum_k \min(\mathbf{z}_i[k], \mathbf{z}^{\text{tgt}}[k])}{\sum_k \max(\mathbf{z}_i[k], \mathbf{z}^{\text{tgt}}[k])}, \tag{9}$$

where $\mathbf{z}_i[k]$ (or $\mathbf{z}^{\text{tgt}}[k]$) denotes the $k$-th element of the corresponding embedding.

## 3   Experiments

In this section, we design experiments to systematically evaluate our method (MONA) for task-specific instruction tuning. We center our evaluation around the following key questions:

- **Effectiveness and Robustness (Q1)**: Does MONA consistently select data that yields better downstream performance across (i) various source general instruction datasets and target evaluation tasks, (ii) different instruction-tuned LLMs, and (iii) a range of data selection ratios?

---

[8]We provide an ablation study of the effect of different $K$ values in § 3.5.

Table 1: Performance of different models after instruction tuning with **5%** of the data selected from different datasets. **Best** results are in bold; second best are underlined.

| Method | $\mathcal{D}^{\text{src}}$ =OpenHermes-2.5 | | | | | | $\mathcal{D}^{\text{src}}$ =Less | | | |
|---|---|---|---|---|---|---|---|---|---|---|
| | MMLU | GSM8K | BBH | MBPP | GPQA | Avg. | MMLU | BBH | TydiQA | Avg. |
| *LLaMA3.1-8B* | | | | | | | | | | |
| BASE | 65.30 | 55.50 | 63.08 | 46.40 | 28.12 | 51.68 | 65.30 | 63.08 | 71.26 | 66.55 |
| FULL | 64.60 | 65.35 | 64.31 | 49.00 | 27.90 | 54.23 | 64.60 | 64.31 | 72.66 | 67.19 |
| RANDOM | 64.02 | 58.65 | 63.70 | 46.73 | 30.36 | 52.69 | 64.16 | **64.29** | 69.78 | 66.08 |
| *Influence-based* | | | | | | | | | | |
| MATES | 64.11 | 54.28 | 65.38 | 47.60 | 28.12 | 51.90 | 63.62 | 63.68 | 67.74 | 65.01 |
| LESS | 64.34 | 66.87 | 63.00 | 47.80 | **31.47** | 54.70 | 62.51 | 62.11 | 70.68 | 65.10 |
| *Distribution alignment* | | | | | | | | | | |
| BM25 | 64.14 | 66.64 | 65.23 | 48.40 | 27.90 | 54.46 | 64.41 | 63.74 | 68.07 | 65.41 |
| DSIR | 63.95 | 66.94 | 64.29 | 48.60 | 29.91 | 54.74 | 64.25 | 63.19 | 65.61 | 64.35 |
| DLRDS-BGE | 64.45 | 64.82 | 64.20 | 48.60 | 31.25 | 54.66 | 64.06 | 61.82 | 70.30 | 65.39 |
| DLRDS-LLaMA3-8B | 64.31 | 64.75 | 63.97 | **48.80** | 29.46 | 54.26 | 62.11 | 61.54 | 71.91 | 65.19 |
| LLM2Vec | 64.29 | 63.53 | 65.55 | 48.40 | 30.13 | 54.38 | 62.06 | 62.03 | 68.11 | 64.07 |
| MoNA (ours) | **64.49** | **67.93** | **66.44** | 48.40 | **31.47** | **55.75** | **64.78** | 64.21 | **72.60** | **67.20** |
| *OLMo-7B* | | | | | | | | | | |
| BASE | 28.42 | 7.35 | 29.96 | 21.40 | 26.56 | 22.74 | 28.42 | 29.96 | 31.67 | 30.02 |
| FULL | 45.05 | 31.96 | 33.13 | 26.40 | 26.56 | 32.62 | 39.31 | 28.86 | 33.43 | 33.87 |
| RANDOM | 36.96 | 16.00 | 31.47 | 19.47 | 27.38 | 26.26 | 28.60 | 30.82 | 31.93 | 30.45 |
| *Influence-based* | | | | | | | | | | |
| MATES | 30.27 | 13.72 | 32.33 | 16.40 | 27.01 | 23.95 | 29.57 | 30.46 | 31.02 | 30.35 |
| LESS | **46.15** | 26.91 | 33.68 | 20.20 | 25.89 | 30.57 | 37.21 | 30.07 | 33.20 | 33.49 |
| *Distribution alignment* | | | | | | | | | | |
| BM25 | 42.34 | 31.08 | **34.30** | **26.80** | 25.45 | 31.99 | 35.74 | 28.95 | **34.40** | 33.03 |
| DSIR | 36.48 | 29.26 | 34.08 | 19.40 | 27.23 | 29.29 | 29.54 | **32.87** | 33.25 | 31.89 |
| DLRDS-BGE | 42.77 | 32.30 | 33.40 | **26.80** | 23.88 | 31.83 | 35.22 | 25.65 | 33.28 | 31.38 |
| DLRDS-LLaMA3-8B | 38.16 | 31.39 | 33.30 | 22.80 | **30.13** | 31.16 | **40.64** | 26.08 | 31.08 | 32.60 |
| LLM2Vec | 37.24 | 30.10 | 33.57 | 23.40 | 28.35 | 30.53 | 39.72 | 28.58 | 32.26 | 33.52 |
| MoNA (ours) | 44.74 | **32.83** | 33.51 | 26.00 | 25.00 | **32.42** | 40.14 | 30.19 | 33.80 | **34.71** |

- **Visualization and Interpretability (Q2)**: Can the monosemantic activation embeddings make data selection decisions more transparent and interpretable? We illustrate this via visual analysis of activation patterns.

- **Key Factor Analysis (Q3)**: How do crucial factors—such as layer selection, sparsity parameter $K$, and similarity metric—affect the behavior of MoNA?

## 3.1  Experimental Setup

**General Instruction Data and Evaluation Tasks**   To comprehensively evaluate robustness and generalization on target tasks, we select training data from two large-scale, diverse instruction datasets: OpenHermes-2.5 [9] (1M synthetic and curated instruction/chat samples) and Less [11] (270K samples covering both classical sources such as Flan V2 [8], CoT [34], and open-ended human-annotated datasets like Dolly [35] and Open Assistant 1 [36]). We evaluate performance on six target tasks: MMLU [37] (general knowledge), BBH [38] (complex reasoning), GSM8K [39] (math problems), MBPP [40] (programming), GPQA [41] (expert QA), and TydiQA [42] (multilingual QA). Evaluations use lm-evaluation-harness [43] and vLLM [44] except for TydiQA, which uses the LESS codebase [11]. More details are in Appendix B.1.

**Models and Training**   We conduct instruction tuning on two widely used open-source language models: LLaMA3.1-8B [5] and OLMo-7B [45], with additional experiments on a larger model, LLaMA2-13B [46], to assess scalability. For data selection, we utilize open-source sparse autoencoder (SAE) models based on LLaMA3-8B[9], with setting $K = 192$ (§ 2.3). Neuron activations are extracted from the penultimate (second-to-last) layer of the model. Fine-tuning is performed with llama-factory [47], using a cosine scheduler (peak learning rate $7e-6$, warmup ratio $0.01$), batch size 128, weight decay 0.1, and maximum sequence length 8192. All models are trained for two epochs. More details are shown in Appendix A.

---

[9]https://huggingface.co/EleutherAI/sae-llama-3-8b-32x

Table 2: Performance of LLaMA2-13B after instruction tuning with **5%** of the data selected from OPENHERMES-2.5. **Best** results are in bold; second best are underlined.

| Method | MMLU | GSM8K | BBH | MBPP | GPQA | Avg. |
|---|---|---|---|---|---|---|
| BASE | 55.11 | 24.03 | 46.74 | 27.00 | 30.58 | 36.69 |
| FULL | 57.61 | 55.95 | 52.63 | 35.00 | 27.01 | 45.64 |
| RANDOM | 55.96 | 41.50 | 51.37 | 31.13 | **28.64** | 41.72 |
| *Influence-based* | | | | | | |
| MATES | 55.68 | 37.07 | 51.17 | 31.20 | 26.34 | 40.29 |
| LESS | **60.38** | 48.75 | 50.42 | 25.80 | 27.90 | 42.65 |
| *Distribution alignment* | | | | | | |
| BM25 | 57.60 | 58.15 | **52.65** | 34.60 | 27.90 | 46.18 |
| DSIR | 55.83 | 53.53 | 52.02 | 31.60 | 27.23 | 44.04 |
| DLRDS-BGE | 56.65 | 56.63 | 52.34 | **35.60** | 27.68 | 45.78 |
| DLRDS-LLaMA3-8B | 58.65 | 52.31 | 52.36 | **35.20** | 26.56 | 45.02 |
| LLM2Vec | 57.02 | 58.30 | 51.27 | 34.80 | 27.90 | 45.86 |
| MoNA (ours) | 57.26 | **60.27** | 52.23 | **35.60** | 27.90 | **46.65** |

## 3.2 Baselines

To ensure fair and comprehensive comparison, we evaluate MONA against several representative baselines covering all major categories from § 1: (i) *Non-selection baselines*: BASE (base pretrained model) and FULL (instruction tuning on full instruction data); (ii) *Random selection* (RANDOM): uniformly samples data for fine-tuning (averaged over three seeds); (iii) *Influence-based selection*: MATES [13] (proxy model predicts loss reduction per sample) and LESS [11] (gradient-based Taylor approximation estimates influence); (iv) *Distribution alignment-based selection*: includes classical methods such as BM25 [18] (tf-idf) and DSIR [19] (n-gram), as well as deep embedding approaches— DLRDS-BGE [20] (BGE embeddings), DLRDS-LLaMA3-8B [5] (LLaMA3-8B embeddings), and LLM2Vec [21] (bidirectional text encoders adapted from decoder-only LLMs). All baselines use the same data selection ratio for fair comparison; additional details are provided in Appendix C.

## 3.3 Main Results on Target Tasks

To address Q1 (effectiveness and robustness), we select 5% of the general instruction data for each data selection method, fine-tune two backbone models, and evaluate on target tasks. We also assess scalability by repeating the experiments with a larger model. In addition, we investigate the impact of varying the data selection ratio, and further employ an LLM-based data analyst to conduct a model-agnostic evaluation of the selected data quality.

**Across Datasets and Target Tasks** As shown in Table 1, on the LLaMA3.1-8B model, MONA achieves either the best or second-best performance on nearly all tasks for both OPENHERMES-2.5 and LESS instruction datasets. For instance, on OPENHERMES-2.5, MONA achieves the highest scores on GSM8K, BBH, and GPQA, and obtains the best overall average—even surpassing full-data fine-tuning. Similar trends are observed on LESS. These results demonstrate that MONA not only selects more semantically relevant data than all baselines, but also maintains robust performance across a wide range of tasks and instruction data sources.

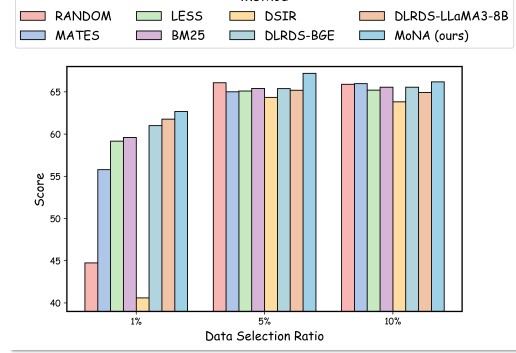

Figure 3: Performance of different data selection methods under varying selection ratios, evaluated on LESS with LLaMA3.1-8B.

**Across Backbone Models** On OLMo-7B, MONA achieves the highest overall average performance among all methods for both OPENHERMES-2.5 and LESS instruction datasets (Table 1). Although task-level stability[10] decreases for all methods compared to LLaMA3.1-8B, MONA still achieves the highest proportion of top-two finishes—ranking in the top two on 3 out of 5 tasks for

---

[10]Here, task-level stability refers to the proportion of tasks where a method ranks among the top two. Lower stability means high performance is achieved on fewer tasks.

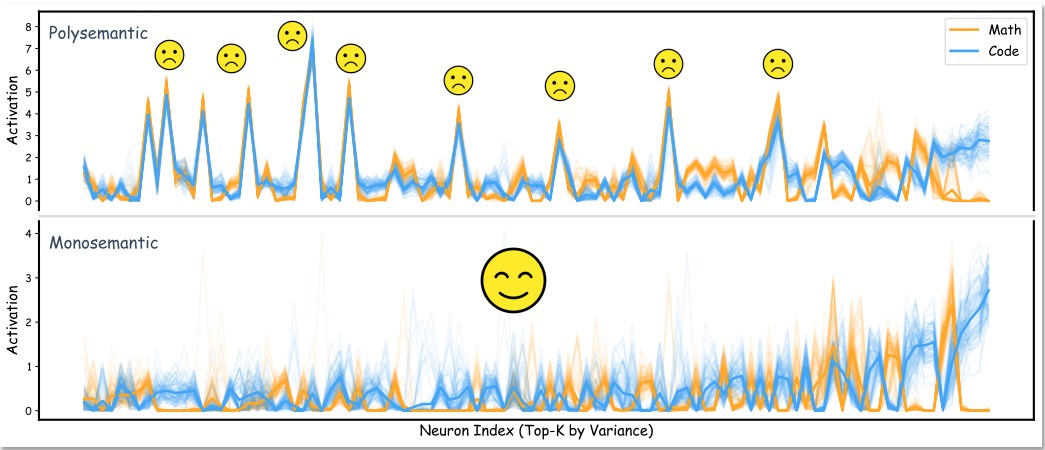

Figure 4: Neuron activation profiles for 100 Math and 100 Code samples on the top-100 most variant neurons. Faint lines show individual samples; bold lines show task means. In the polysemantic (top) plot, many neurons, especially those with high activation peaks (marked by weeping face), are simultaneously activated by both tasks, reflecting pronounced overlap and limited task specificity. In contrast, the monosemantic (bottom) plot reveals clear task-specific activation patterns.

OPENHERMES-2.5 and 2 out of 3 tasks for LESS, both of which are higher than any baseline. This indicates that, despite the absolute stability being affected by the backbone, MONA remains the most robust and semantically expressive data selection approach relative to competing methods.

On LLaMA2-13B, MONA exhibits a similar trend as observed on OLMo-7B (Table 2). Although absolute task-level stability decreases compared to LLaMA3.1-8B, MONA continues to show stronger overall performance and relatively higher stability than all baseline methods.

**Across Data Selection Ratios**  Across all selection ratios (Figure 3), MONA consistently achieves the best performance, reaffirming both its robustness and semantic expressiveness. Interestingly, selecting 10% of the data results in lower performance than using 5%. We speculate that increasing the ratio may introduce less relevant or lower-quality samples, thereby diluting the benefits of high-quality, semantically aligned data. This observation indicates that the choice of selection ratio is a critical factor in data selection for instruction tuning and deserves further exploration.

**LLM as a Data Analyst**  Beyond evaluating instruction-tuned model performance, we employ an LLM-based data analyst to assess the quality of selected training data in a model-agnostic way. For each method, we randomly sample 100 training instances and prompt GPT-4o-mini [48] to compare them with representative target samples, considering semantic similarity, instruction format, and task relevance. Each comparison is scored from 1 to 10, and final scores are averaged. More details are given in Appendix B.2. As shown in Figure 5, the LLM consistently assigns higher scores to data selected by MONA versus two strong baselines, confirming both the semantic expressiveness and stability of our approach. Additional case studies are provided in Appendix D.1.

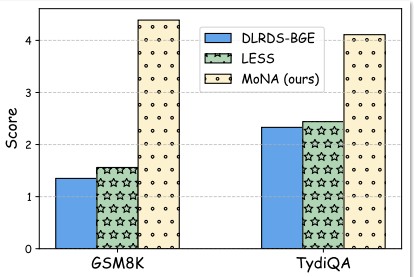

Figure 5: LLM as a Data Analyst: scores for data selected by different methods. Higher scores indicate better performance

### 3.4 Neuron Activation Visualization

In addition to validating improved downstream performance with neuron activation-based data selection, we further analyze and visualize the neuron activation patterns (Q2) for different tasks (Figure 4). While polysemantic neurons produce substantial activation overlap across tasks, monosemantic representations obtained via the sparse autoencoder yield well-separated task-specific activation patterns. This underscores the importance of disentangling polysemantic activations.

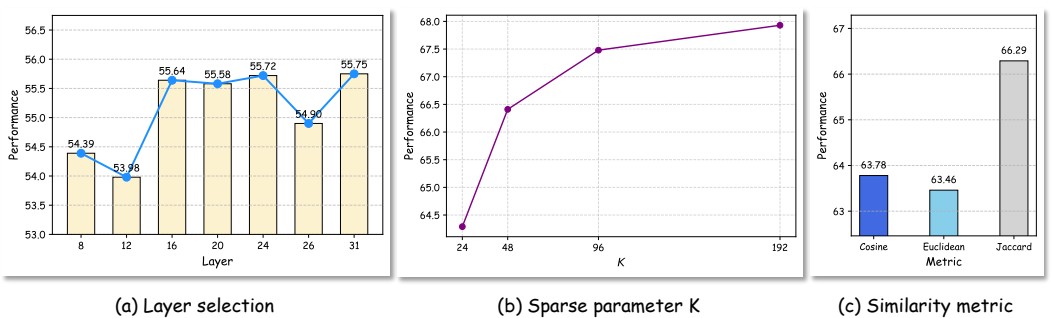

| (a) Layer selection | (b) Sparse parameter K | (c) Similarity metric |

Figure 6: Ablation studies for key design choices in MONA

## 3.5 Ablation Studies

To better understand the contribution of each component in MONA (Q3), we conduct ablation studies on key design choices, including which layer to extract neuron activations from, the sparsity parameter $K$, and the similarity metric. More experimental details (e.g., evaluation benchmarks and instruction tuning datasets) are in Appendix D.8.

**Effect of Layer Selection**   We first examine how the choice of layer for extracting neuron activations affects the performance of MONA. Since prior work [30] shows that SAEs trained on shallower layers tend to specialize in next-token prediction and provide less transferable features, we focus our analysis on deeper layers. Specifically, we select seven layers evenly spaced from layer 8 to the penultimate layer (layer 31) of LLaMA3-8B. As shown in Figure 6-a, embeddings from shallower layers can result in unstable or suboptimal performance, while embeddings from deeper layers—especially the penultimate layer—deliver strong results. Based on these observations, we extract neuron activations from the penultimate layer in all other experiments.

**Effect of Sparsity Parameter $K$**   As shown in Figure 6-b, model performance generally improves as the sparsity parameter $K$ increases, indicating that retaining more active neurons leads to higher-quality data selection. The improvement becomes less pronounced as $K$ grows larger. In contrast, when $K$ is very small, performance drops sharply, suggesting that insufficient neuron information hampers effective selection. Based on these results, we adopt $K = 192$ in all main experiments.[11]

**Effect of Similarity Metric**   We compare the impact of different similarity metrics, including Jaccard, Cosine, and Euclidean. As shown in Figure 6-c, Jaccard similarity consistently yields better results than the other metrics across benchmarks, highlighting its suitability for MONA.

## 3.6 Additional Analysis

**Effect of SAE**   To examine the effect of the SAE model in greater depth, we additionally train an SAE model built upon LLaMA2-13B. The model is trained using the RedPajama-Data-1T-Sample dataset[12] , with training conducted under the sparsify framework[13] . As is shown in Figure 7, a larger SAE backbone can further enhance the effectiveness of data selection and downstream performance. These findings suggest that the quality and scale of the SAE model have a positive impact on the overall results.

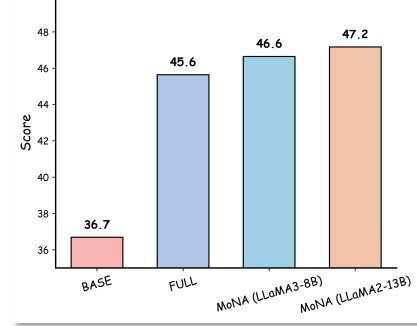

Figure 7: Results of Using Different SAE Models on LLaMA2-13B

**Comparison with MoE-based Monosemantic Embeddings**   To further explore the concept of monosemantic embeddings, we investigate an MoE-based monosemantic embedding approach [49].   Specifically, we replace our SAE with MoE-based

---

[11]We did not explore values of $K$ greater than 192 in this work.

[12]https://huggingface.co/datasets/togethercomputer/RedPajama-Data-1T-Sample

[13]https://github.com/EleutherAI/sparsify/tree/main

embeddings: for each token, we use the concatenation of expert routing scores from all layers of an MoE model (collected during a forward pass) as the token's embedding. For this experiment, we employ the allenai/OLMoE-1B-7B-0924 [50] model. Except for substituting the SAE-based embeddings with MoE-based embeddings, all other components and settings of our workflow remain unchanged to ensure a fair comparison.

Figure 8 demonstrates that using MoE-based embeddings yields a modest improvement over the FULL baseline under our current experimental setup, indicating that monosemantic representations from MoE models can provide a slight benefit for task-specific data selection. However, MoE-based embeddings still lag behind SAE-based embeddings by a noticeable margin. This observation suggests that the MoE-based embedding still has room for improvement in capturing monosemantic representations.

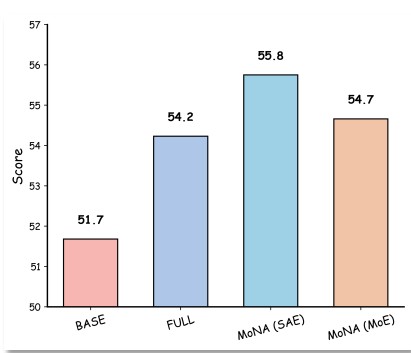

Figure 8: Evaluation of Monosemantic Embeddings over LLaMA3.1-8B: SAE-based vs. MoE-based

## 4 Related Work

**Data Selection for Task-Specific Instruction Tuning** Most data selection methods for task-specific instruction tuning roughly fall into two main categories: influence-based and distribution alignment approaches. Influence-based methods select training data by estimating the influence of each candidate sample on the evaluation loss for target examples [14], and prioritize samples expected to most reduce this loss. For example, DsDm [12] and MATES [13] use proxy models, while LESS [11] relies on gradients. However, these methods can be unstable because evaluation loss may not reliably indicate model quality after instruction tuning [15–17]. Distribution alignment methods [18–21] select samples by embedding them into a feature space and aligning the source and target distributions using a similarity metric. Prior work is largely data-centric, representing samples with surface-level features such as n-grams [19], tf-idf [18], or neural embeddings [20], which may fail to reflect how the model internally processes information [26, 27]. In contrast, our approach is model-centric: we represent each sample by the neuronal activation pattern it triggers inside the model, thereby capturing the internal computational dynamics that are critical for task-specific performance.

**Sparse Autoencoders in Feature Learning** Sparse coding was first introduced for over-complete dictionaries [51], and unsupervised dictionary learning was pioneered by [52]. These ideas led to sparse autoencoders, which have become important tools for learning structured features in vision and language [53, 54]. Recent work has also investigated sparse autoencoders in analyzing representations of large language models [26, 55]. Almost concurrently, [56] used sparse autoencoders for data selection, with a focus on diversity. In contrast, our approach is distinct in that we use sparse neuron activations to capture semantic relatedness between samples and to overcome neuron polysemanticity, enabling more interpretable and semantically aligned task-specific data selection.

## 5 Conclusion

We propose a model-centric approach for task-specific data selection in instruction tuning, using monosemantic neuronal activations from sparse autoencoders. This representation captures internal model computation and enables more semantically aligned and interpretable data selection. Experiments across various models and tasks demonstrate consistent gains over previous baselines.

## Acknowledgments

This work was supported by the China NSFC Projects (92370206, U23B2057, 62120106006), and the Shanghai Municipal Science and Technology Projects (2021SHZDZX0102 and 25X010202846).

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

## Impacts and Limitations

Our proposed data selection framework MONA offers a neuroscience-inspired perspective for improving task-specific instruction tuning. By enhancing the stability, semantic expressiveness, and interpretability of data selection, MONA has the potential to benefit a broad range of downstream language model applications and may inspire new research directions in data curation and model analysis, both in academia and industry.

While MONA demonstrates strong effectiveness for task-specific instruction tuning, its extension to other stages—such as pre-training data selection—as not been explored in this work. In addition, our current study focuses solely on the text modality. Applying MONA to multimodal data selection scenarios, for example image-text tasks, remains an open and promising direction for future research.

## A    Training Details

We provide details on the input data formatting for instruction tuning across the three backbone models evaluated in our work: LLaMA3.1-8B, OLMo-7B, and LLaMA2-13B. For each model, we adopted the format recommended in its official documentation or open-source release. Below, we present a concrete data example for each model. Additionally, all experiments are conducted on NVIDIA A100, A800, and H800 GPUs.

| LLaMA3.1-8B |
|---|
| <\|begin_of_text\|><\|start_header_id\|>user<\|end_header_id\|>

A doctor gives you three pills. She tells you to take one every half hour. How long will the pills last?<\|eot_id\|><\|start_header_id\|>assistant<\|end_header_id\|>

One hour. You take the first pill immediately, then the other two at half-hour intervals.<\|eot_id\|> |

| OLMo-7B |
|---|
| A doctor gives you three pills. She tells you to take one every half hour. How long will the pills last? One hour. You take the first pill immediately, then the other two at half-hour intervals.<\|endoftext\|> |

| LLaMA2-13B |
|---|
| [INST] A doctor gives you three pills. She tells you to take one every half hour. How long will the pills last? [/INST] One hour. You take the first pill immediately, then the other two at half-hour intervals.  |

## B    Evaluation Details

### B.1    Evaluation Tasks Details

For MMLU and MBPP, we directly use their respective validation sets as representative examples. For the remaining datasets without validation sets, we follow the strategies outlined below:

- GSM8K: We randomly select 100 samples from the training set to serve as representative examples.
- BBH: We extract representative examples by selecting the provided few-shot samples in the task setup.
- GPQA: We use the extended 98 data points, which are the "extended split" minus the "main split."
- TydiQA: Following [11], we select one sample per language as the representative example.

Table 3: Details of all evaluation tasks

| Task | $|\mathcal{D}^{\text{tgt}}|$ | # Test Samples | Shot | Metric | Harness Task Name |
|------|------|------|------|------|------|
| MMLU | 285 | $18,721$ | 5 | Accuracy | mmlu |
| GSM8K | 100 | $2,638$ | 8 | Exact Match | gsm8k_cot |
| BBH | 81 | 920 | 3 | Exact Match | bbh |
| MBPP | 90 | 500 | 3 | pass@1 | mbpp |
| GPQA | 98 | 448 | 0 | Accuracy | gpqa_main_zeroshot |
| TydiQA | 9 | $5,077$ | 1 | F1 | - |

For more statistical information, evaluation metrics, and details regarding the LM-evaluation-harness setup, please refer to Table 3.

---

**Prompt of LLM data analyst**

You are an expert in evaluating task-specific data selection strategies for instruction tuning of large language models (LLMs). Your task is to assess how effectively the selected training data improve the performance of LLMs on a specific target task.

### Context:
1. You are provided with a **representative example** of the target task.
2. Only a small sample of the selected data is provided as a reference due to space limitations.
3. Your goal is to evaluate how well the selected data would help fine-tune the LLM to enhance its performance on the example of the target task. You will accomplish this by scoring the model's performance on the target task after being fine-tuned with the provided sample data.

### Instructions:
1. Consider how well the sampled training data aligns with the example in terms of:
   - Semantic similarity: How similar are the contents or instructions to the target task example?
   - Instruction format compatibility: Are the input-output structures of the selected data compatible with the target task?
   - Potential for improving generalization to this target task: How much does the sampled data appear to address challenges in the target task?
2. Rate the effectiveness of the training data on a scale of 1 to 10 for the target task example, where:
   - 1 means the training data is completely irrelevant or harmful.
   - 10 means the training data is highly relevant and likely to maximize performance.
3. Provide a short explanation for your rating.

### Representative Example:
{}

### Sampled Training Data:
{}

### Output:
For the given target task example and sampled training data, provide the following:
1. A rating (1-10) for the sampled data based on the criteria above.
2. A brief explanation justifying your rating.

---

## B.2    LLM Data Analyst Details

We select GPT-4o-mini [48] as the LLM data analyst. The prompt used is shown above. When calling the API, the temperature is set to $0.8$.

### B.3 Polysemantic Neuronal Activation Extraction Details

We adopt the method from CATS [57] to extract polysemantic neuronal activations from a specified layer of the large language model, specifically within the Gated-MLP block. Mathematically, let $\mathbf{x} \in \mathbb{R}^{d_1}$ denote the input to the Gated-MLP. The polysemantic neuron activation pattern is computed as

$$\mathbf{z}^{\text{poly}} = TopK\left(\left|SiLU\left(\mathbf{W}_{\text{gate}}\mathbf{x}\right)\right|\right), \tag{10}$$

where $\mathbf{W}_{\text{gate}}$ is the learnable weight matrix in the block, and $SiLU\left(\cdot\right)$ is the activation function [58]. The operator $TopK(\cdot)$ retains only the largest $K$ values of the input vector and sets all remaining entries to zero, consistent with the extraction procedure for the monosemantic activations.

To ensure a fair comparison, both polysemantic and monosemantic activations are extracted from the same layer and with the same value of $K$, as used in the results of Figure 1-c and Figure 4.

## C  Baseline Details

### C.1  MATES

We implement MATES based on the official repository[14]. Similar to the original work, we trained a data model based on `bert-base-uncased` [59] with a maximum sequence length of 4096. The batch size is 32, the learning rate is 5e-5, and we trained for 5 epochs with a weight decay of 0.01. All other hyperparameters were kept consistent with the original work.

### C.2  LESS

The experiments were run using the official repository[15]. Apart from the hyperparameters mentioned in § 3.1, all other parameters remain consistent with the original work.

### C.3  DLRDS

For this method, we considered two data selection models: `bge-base-en-v1.5` and `LLaMA3-8B`. In Appendix D.2, since the experiments were conducted on Chinese data, we used `bge-base-zh`. When selecting data with this method, we applied the Cosine similarity metric. When using `bge-base-en-v1.5` or `bge-base-zh`, if a sentence exceeded the model maximum token length (e.g., 512), we split the sentence into non-overlapping chunks that fit within the length limit. The embedding for the entire sentence was computed as the average of the embeddings of all chunks.

### C.4  BM25

To ensure speed, we used the `bm25s` [60] toolkit[16].

### C.5  DSIR

We implement based on the official repository[17].

### C.6  LLM2Vec

We used the official open-source implementation[18] and the released model `LLM2Vec-Sheared-LLaMA-mntp` for all experiments with this method.

## D  Additional Experimental Results and Analysis

### D.1  Case Study: LLM as a Data Analyst

Figure 5 shows that, compared to other data selection methods, the LLM data analyst consistently finds that our approach has a clear advantage. To further analyze why MONA receives higher scores,

---

[14]https://github.com/cxcscmu/MATES
[15]https://github.com/princeton-nlp/LESS
[16]https://github.com/xhluca/bm25s
[17]https://github.com/p-lambda/dsir
[18]https://github.com/McGill-NLP/llm2vec?tab=readme-ov-file

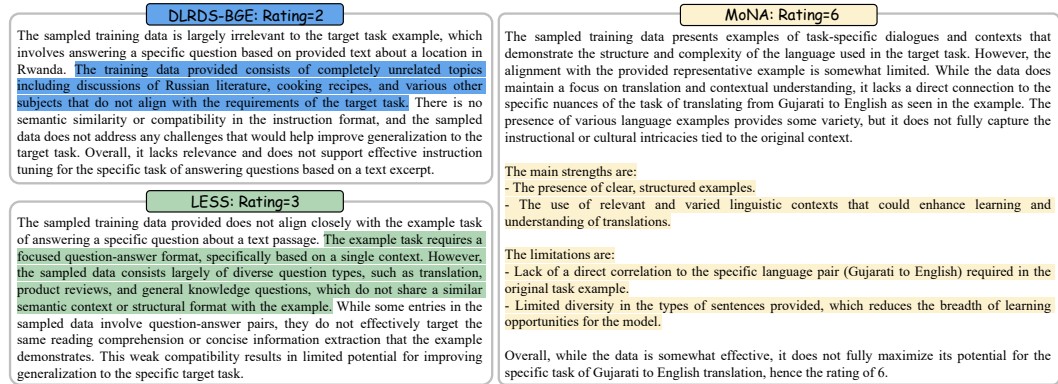

Figure 9: Explanation of the score given by the LLM data analyst for data selected by different methods

Table 4: Recall rate of the knowledge required for physical questions across different methods

| Method | DLRDS-BGE | LESS | MONA (ours) |
|--------|-----------|------|-------------|
| Hit@10 | 78.69 | 13.93 | **81.15** |

we provide a case study in Figure 9, which illustrates the LLM analyst evaluation for the same TydiQA example selected by different methods. In this case, DLRDS-BGE receives a low score due to selecting irrelevant topics, while LESS is penalized for choosing data that does not match the question-answer format. These low scores are justified, since empirical evidence indicates that semantic relevance and format alignment between training and test data are critical for downstream performance. In contrast, MONA selects data satisfying both criteria, even though the samples are in different language pairs. The LLM data analyst considers this selection acceptable, likely because semantically similar content across languages can also improve model performance, as supported by previous work [61, 62].

## D.2 Knowledge Recall for Domain-Specific Questions

To further evaluate data selection quality on specialized tasks, we conduct a knowledge recall experiment using the physics subset from [63]. Each of the 122 physics questions covers specific knowledge points, with a total of 84 distinct points represented. For each data selection method, we measure the recall rate of relevant knowledge, that is, the proportion of target knowledge points that are present in the selected training data. As reported in Table 4, MONA achieves higher recall rates compared to DLRDS-BGE, while LESS demonstrates a low recall success rate. This difference likely arises from a greater mismatch between the distribution of knowledge data and question data for some methods, leading to larger gradient discrepancies and less effective selection. These results indicate that MONA is more effective in retrieving domain-specific knowledge essential for answering specialized questions.

## D.3 Effect of Data Relevance on Fine-Tuning Performance

To assess the impact of data relevance, we conduct experiments comparing fine-tuning performance using data that is either highly similar or highly dissimilar to the target task, as determined by Jaccard similarity. Specifically, we select subsets of training samples that are the most similar or most dissimilar to the target task, and use each subset for instruction tuning. As shown in Table 5, using dissimilar data for fine-tuning leads to a significant reduction in performance across benchmarks, especially on GSM8K, whereas using similar (relevant) data consistently yields better results. These findings underscore the importance of selecting task-relevant data and validate the effectiveness of Jaccard similarity in our data selection framework.

Table 5: Performance comparison of fine-tuning with data dissimilar versus similar to the target task (measured by Jaccard similarity, $\mathcal{D}^{\text{src}} = \text{OPENHERMES-2.5}$)

| Selection | MMLU | GSM8K | BBH | Avg. |
|---|---|---|---|---|
| dissimilar | 63.38 | 33.06 | 61.17 | 52.54 |
| similar | 64.49 | 67.93 | 66.44 | 66.29 |

Table 6: Performance of different data curation methods on TydiQA ($\mathcal{D}^{\text{src}} = \text{LESS}$)

| Method | Data Curation Paradigm | TydiQA |
|---|---|---|
| GPT-4o (Self-instruct) | data synthesis | 46.99 |
| MoNA (ours) | data selection | 72.60 |

## D.4 Comparison with Self-Instruct Data Synthesis

To further compare data curation paradigms, we evaluate MoNA against a GPT-4o-based self-instruct method [64][19] that synthesizes instruction data (data synthesis), as commonly adopted in recent literature. Table 6 reports the TydiQA performance for both approaches. MoNA (data selection) significantly outperforms GPT-4o (self-instruct), achieving a score of 72.60 versus 46.99. This disparity can be attributed to the tendency of self-instruct methods to generate large amounts of data that may become increasingly redundant and less diverse as the volume grows. In contrast, MoNA leverages data selection to curate a more relevant and diverse subset from an existing dataset. When a sufficiently large pool of real data is available, selecting high-quality samples may be a more effective and straightforward approach than synthesizing new data.

## D.5 Analysis of Length Bias in Token Aggregation

**Empirical Analysis** Table 7 presents the performance and average selected sample length for random selection, MoNA without length normalization, and MoNA with length normalization across multiple tasks. When sum aggregation is used (i.e., without normalization), MoNA tends to select samples whose lengths are closest to the average length of the representative target samples, rather than samples with the highest semantic relevance, and model performance suffers as a result. In contrast, applying length normalization (mean aggregation) effectively removes this bias, enabling the selection of samples with a broader range of lengths and higher semantic utility. This results in significant improvements in downstream task performance. These empirical findings demonstrate that length normalization in token aggregation is essential for mitigating length bias and achieving more reliable data selection for MoNA.

**Theoretical Analysis** Below, we show why sum aggregation induces a length bias, and how mean aggregation mitigates it. Let each sample $\mathbf{s}_i$ consist of $n_i$ tokens, each mapped to a non-negative $d$-dimensional activation vector $\mathbf{z}_{i,j}$. Under sum aggregation, the sample embedding is

$$\mathbf{z}_i = \sum_{j=1}^{n_i} \mathbf{z}_{i,j}. \tag{11}$$

We use the generalized Jaccard similarity between $\mathbf{z}_i$ and the task prototype $\mathbf{z}^{\text{tgt}}$, defined as:

$$\mathcal{J}\left(\mathbf{z}_i, \mathbf{z}^{\text{tgt}}\right) = \frac{\sum_k \min(\mathbf{z}_i[k], \mathbf{z}^{\text{tgt}}[k])}{\sum_k \max(\mathbf{z}_i[k], \mathbf{z}^{\text{tgt}}[k])}. \tag{12}$$

In sum aggregation, the magnitude of $\mathbf{z}_i[k]$ scales linearly with the number of tokens $n_i$, since $\mathbf{z}_i[k] \approx m_{i,k} \cdot v_k$, where $m_{i,k}$ is the number of tokens in $\mathbf{s}_i$ that activate dimension $k$, and $v_k$ is the typical activation value. Similarly, $\mathbf{z}^{\text{tgt}}[k]$ reflects the activation strength of the target prototype, which is influenced by the average length of representative samples of the target task. When the length $n_i$ of $\mathbf{s}_i$ is similar to the length associated with $\mathbf{z}^{\text{tgt}}$, the magnitudes of $\mathbf{z}_i[k]$ and $\mathbf{z}^{\text{tgt}}[k]$ are comparable. If their activated dimensions overlap well, the Jaccard similarity is maximized. However:

---

[19] https://github.com/yizhongw/self-instruct

Table 7: Performance and average selected sample length for RANDOM, MoNA w/o length normalization, and MoNA w/ length normalization (LN) across tasks ($\mathcal{D}^{\text{src}} = $ OPENHERMES-2.5). The "Target Length" column shows the average length of representative target samples.

| Task | Target Length | RANDOM | | MoNA **w/o LN** | | MoNA **w/ LN** | |
|---|---|---|---|---|---|---|---|
| | | *Performance* | *Length* | *Performance* | *Length* | *Performance* | *Length* |
| MMLU | 150.85 | 64.39 | | 62.88 | 152.76 | 64.49 | 499.41 |
| GSM8K | 189.66 | 60.05 | | 64.06 | 212.70 | 67.93 | 322.74 |
| BBH | 303.49 | 64.63 | 391.34 | 65.55 | 313.82 | 66.44 | 492.14 |
| MBPP | 110.38 | 49.67 | | 46.20 | 202.67 | 48.40 | 404.03 |
| GPQA | 370.54 | 30.43 | | 27.00 | 309.88 | 31.47 | 632.61 |

Table 8: Performance comparison using LESS data: polysemantic vs. monosemantic activations

| Neuronal Activation | MMLU | BBH | TydiQA | Avg. |
|---|---|---|---|---|
| polysemantic | 63.57 | 63.28 | 71.01 | 65.95 |
| monosemantic | 64.78 | 64.21 | 72.60 | 67.20 |
| w/o SAE | 62.11 | 61.54 | 71.91 | 65.19 |

- If $n_i$ is much larger than the effective length of $\mathbf{z}^{\text{tgt}}$, then $\mathbf{z}_i[k] \gg \mathbf{z}^{\text{tgt}}[k]$ for many dimensions $k$, leading to $\min(\mathbf{z}_i[k], \mathbf{z}^{\text{tgt}}[k]) = \mathbf{z}^{\text{tgt}}[k]$ and $\max(\mathbf{z}_i[k], \mathbf{z}^{\text{tgt}}[k]) = \mathbf{z}_i[k]$. Consequently, $\mathcal{J}(\mathbf{z}_i, \mathbf{z}^{\text{tgt}}) \approx \frac{\sum_k \mathbf{z}^{\text{tgt}}[k]}{\sum_k \mathbf{z}_i[k]} \ll 1$, reducing the similarity.

- If $n_i$ is much smaller, $\mathbf{z}_i[k] \ll \mathbf{z}^{\text{tgt}}[k]$, yielding $\mathcal{J}(\mathbf{z}_i, \mathbf{z}^{\text{tgt}}) \approx \frac{\sum_k \mathbf{z}_i[k]}{\sum_k \mathbf{z}^{\text{tgt}}[k]} \ll 1$, again lowering the similarity.

Thus, samples with lengths $n_i$ close to the length of $\mathbf{z}^{\text{tgt}}$ achieve higher similarity scores, introducing a length bias in the selection process.

In contrast, mean aggregation normalizes each sample embedding by the number of tokens. This normalization ensures that the magnitude of the embedding does not depend on the sample length. As a result, the selection process is no longer biased toward samples with lengths similar to the prototype. Instead, comparisons focus purely on the similarity of activation patterns, eliminating the systematic length bias observed with sum aggregation and enabling selection based on semantic relevance.

## D.6 Effect of SAE

Monosemantic activations, produced by the sparse autoencoder, consistently outperform polysemantic activations across all tasks on the LESS dataset (Table 8). This demonstrates that disentangling neuron polysemanticity via SAE leads to more effective data selection and superior downstream performance. Additionally, we report results using the raw hidden states of the selected layer without SAE mapping ("w/o SAE"). This baseline underperforms both polysemantic and monosemantic representations, underscoring the importance of explicit disentanglement with SAE for optimal data selection.

Table 9: Lexical Diversity (MTLD, larger is better) of Selected data

| Method | MMLU | BBH | TydiQA |
|---|---|---|---|
| RANDOM | 61.31 | 61.31 | 61.31 |
| MATES | 68.25 | 56.83 | 81.01 |
| LESS | 76.74 | 53.56 | 77.80 |
| BM25 | 74.89 | 58.41 | 66.65 |
| DSIR | 52.10 | 48.90 | 54.01 |
| DLRDS-BGE | 50.79 | 44.63 | 70.63 |
| DLRDS-LLaMA3-8B | 49.38 | 40.50 | 82.15 |
| MoNA (ours) | 66.05 | 42.49 | 81.38 |

### D.7 Diversity of Selected Data by MONA

We measure the lexical diversity of the selected data samples using the MTLD (Measure of Textual Lexical Diversity) metric [65], computed with the LexicalRichness[20] package. A higher MTLD score indicates greater lexical diversity. Table 9 confirm that a stronger emphasis on data quality in target-specific selection indeed leads to reduced data diversity.

### D.8 Detailed Experimental Results

We present complete results for all tasks and settings discussed in the main text. To facilitate understanding, we summarize key findings in the main text using figures and overview tables, while the following tables provide detailed, task-level results and additional experimental details. These comprehensive tables (10, 11, 12, 13, 14) complement the main text by offering the full performance breakdown, allowing readers to reproduce, verify, or further analyze various aspects of the experiments.

Table 10: Detailed results of Figure 3 ($\mathcal{D}^{\text{src}} = \text{LESS}$)

| Method | 1% | | | | 10% | | | |
|---|---|---|---|---|---|---|---|---|
| | *MMLU* | *BBH* | *TydiQA* | Avg. | *MMLU* | *BBH* | *TydiQA* | Avg. |
| RANDOM | 65.71 | 6.77 | 61.77 | 44.75 | 63.23 | 63.35 | 71.14 | 65.90 |
| MATES | 63.22 | 53.00 | 51.15 | 55.79 | 63.62 | 64.28 | 70.04 | 65.98 |
| LESS | 63.58 | 56.59 | 57.31 | 59.16 | 63.27 | 61.77 | 70.56 | 65.20 |
| BM25 | 63.96 | 47.50 | 67.36 | 59.61 | 63.47 | 62.06 | 71.15 | 65.56 |
| DSIR | 64.91 | 0.20 | 56.73 | 40.61 | 63.17 | 61.07 | 67.22 | 63.82 |
| DLRDS-BGE | 64.71 | 65.58 | 52.74 | 61.01 | 63.45 | 61.59 | 71.61 | 65.55 |
| DLRDS-LLaMA3-8B | 63.48 | 59.53 | 62.31 | 61.77 | 62.28 | 59.91 | 72.57 | 64.92 |
| MONA (ours) | 64.57 | 56.67 | 66.77 | 62.67 | 62.85 | 62.94 | 72.74 | 66.18 |

Table 11: Detailed results of Figure 5

| Method | *GSM8K* ($\mathcal{D}^{\text{src}} = \text{OPENHERMES-2.5}$) | *TydiQA* ($\mathcal{D}^{\text{src}} = \text{LESS}$) |
|---|---|---|
| DLRDS-BGE | 1.35 | 2.33 |
| LESS | 1.56 | 2.44 |
| MONA (ours) | 4.39 | 4.11 |

Table 12: Detailed results of Figure 6-(a) ($\mathcal{D}^{\text{src}} = \text{OPENHERMES-2.5}$)

| Layer | *MMLU* | *GSM8K* | *BBH* | *MBPP* | *GPQA* | Avg. |
|---|---|---|---|---|---|---|
| 8 | 63.94 | 63.15 | 64.75 | 48.20 | 31.92 | 54.39 |
| 12 | 64.06 | 63.46 | 64.75 | 48.40 | 29.24 | 53.98 |
| 16 | 64.73 | 67.02 | 66.47 | 49.20 | 30.80 | 55.64 |
| 20 | 64.36 | 68.54 | 65.21 | 49.00 | 30.80 | 55.58 |
| 24 | 64.13 | 69.52 | 65.27 | 52.00 | 27.68 | 55.72 |
| 26 | 63.11 | 67.10 | 64.06 | 49.20 | 31.03 | 54.90 |
| 31 | 64.49 | 67.93 | 66.44 | 48.40 | 31.47 | 55.75 |

Table 13: Detailed results of Figure 6-b ($\mathcal{D}^{\text{src}} = \text{OPENHERMES-2.5}$)

| $K$ | 192 | 96 | 48 | 24 |
|---|---|---|---|---|
| GSM8K | 67.93 | 67.48 | 66.41 | 64.29 |

## E Algorithm

A complete description of our algorithm is provided (Algorithm 1) in the form of pseudocode, to facilitate reproducibility and implementation in future work.

---

[20]https://github.com/LSYS/LexicalRichness

Table 14: Detailed results of Figure 6-c ($\mathcal{D}^{\text{src}} = $ OPENHERMES-2.5)

| Method | *MMLU* | *GSM8K* | *BBH* | **Avg.** |
|--------|--------|---------|-------|----------|
| Cosine | 63.30 | 66.72 | 61.33 | 63.78 |
| Euclidean | 63.03 | 65.73 | 61.62 | 63.46 |
| Jaccard | 64.49 | 67.93 | 66.44 | 66.29 |

---

**Algorithm 1** MONA: Task-Specific Data Selection with Monosemantic Neuronal Activations

---

**Require:** Source dataset $\mathcal{D}^{\text{src}}$, target set $\mathcal{D}^{\text{tgt}}$, data selection model $\mathcal{M}^{\text{ds}}$, chosen layer $L$, trained SAE, sparsity $K$, selection size $n$
**Ensure:** Selected subset $\mathcal{D}^{\text{sel}} \subset \mathcal{D}^{\text{src}}$ of size $n$
1: **For** each sample **s** in $\mathcal{D}^{\text{src}} \cup \mathcal{D}^{\text{tgt}}$:
2:    Compute monosemantic activation $\mathbf{z}_s$ as in Eq. (6) and aggregate to sample-level as in Eq. (7)
3: Compute target prototype $\mathbf{z}^{\text{tgt}}$ using Eq. (8) on all $\mathbf{z}_j$ in $\mathcal{D}^{\text{tgt}}$
4: **for** each source sample $\mathbf{s}_i$ in $\mathcal{D}^{\text{src}}$ **do**
5:    Compute similarity $s_i$ between $\mathbf{z}_i$ and $\mathbf{z}^{\text{tgt}}$ as in Eq. (9)
6: **end for**
7: Select $\mathcal{D}^{\text{sel}} = $ the $n$ samples in $\mathcal{D}^{\text{src}}$ with highest similarity $s_i$
8: **return** $\mathcal{D}^{\text{sel}}$

---

