# OpenReview forum: "Task-Specific Data Selection for Instruction Tuning via Monosemantic Neuronal Activations"
_NeurIPS.cc/2025/Conference — NeurIPS 2025 poster_

### Official Review · Reviewer_Uiye · 2025-06-07

**Clarity:** 4
**Significance:** 3
**Originality:** 2
**Rating:** 5
**Confidence:** 4

**Summary:**

This paper introduces a novel method for selecting examples for instruction tuning by measuring their similarity to target task examples. The approach leverages the activations induced in a large language model (LLM), selecting examples that elicit similar internal representations. To better align these activations with individual concepts, a sparse autoencoder is employed, encouraging each activation to focus on a distinct concept.

**Questions:**

See in the section above

**Ethical Concerns:**

["NO or VERY MINOR ethics concerns only"]

**Final Justification:**

I have two concerns,
The authors have clarified some misunderstandings I had and conducted a few experiments to validate one of my points.

**Limitations:**

yes

**Quality:**

3

**Strengths And Weaknesses:**

Strengths:

1) The paper is well-written and easy to follow.

2) The proposed method is simple and intuitive—in a positive sense.

3) The experimental setup is thorough, featuring a wide range of strong baselines and diverse, representative datasets.

4) The inclusion of ablation studies provides valuable insight into the components that contribute most to the method's effectiveness.

Suggestions and Questions:

While these may not be strict weaknesses, I have a few suggestions and questions for consideration:

1) In classical active learning literature, diversity among selected examples is often emphasized to ensure broader coverage of the task space. Have you considered strategies to avoid selecting overly similar examples? It would be helpful to see an analysis of task coverage beyond simply demonstrating improved performance.



2) I appreciated the intuition offered in Figure 4, which illustrates that without SEA, it is more challenging to separate activations for distinct tasks. However, do you have a corresponding analysis showing that using SEA leads to improved performance? In Figure 6b, it appears that increasing K generally improves overall performance. Am I missing something here? Is the effect of using "raw" activations explicitly represented in any of the baselines?

---

> ### Author Rebuttal · Authors · 2025-07-30
>
> ## Response to Reviewer Uiye
> Thank you for your valuable suggestions and the time you have dedicated to reviewing our work. Your feedback has been very helpful in improving our research.
>
> ---
>
> ### Clarification on SAE vs. Raw Activations Performance Comparison
> We apologize if our writing led to any important information being overlooked. In fact, we have provided results comparing the use of SAE and raw activations in **Figure 1-(c) and Table 8 in Appendix D.6** of the manuscript. For your convenience, we have reproduced the detailed results below.
>
> **Table 1: Comparison of Different Neuron Activations**
> | Neuronal Activation | MMLU  | BBH   | TydiQA | Avg.  |
> | ------------------- | ----- | ----- | ------ | ----- |
> | polysemantic  [1]   | 63.57 | 63.28 | 71.01  | 65.95 |
> | monosemantic        | 64.78 | 64.21 | 72.60  | 67.20 |
> | w/o SAE             | 62.11 | 61.54 | 71.91  | 65.19 |
>
> Table 1 compares three types of neuronal activations:
> * *Polysemantic*: we adopt the method from CATS [1] to extract polysemantic neuronal activations from a specified layer of the large language model, specifically within the Gated-MLP block. Mathematically, let $x\in\mathbb{R}^{d_1}$ denote the input to the Gated-MLP. The polysemantic neuron activation pattern is computed as:
> $$z_\text{poly}=\text{TopK}(∣\text{SiLU}(W_\text{gate}x)∣),$$
> where $W_{\text{gate}}$ is the learnable weight matrix in the block, and $\text{SiLU}(·)$ is the activation function. The operator $\text{TopK}(·)$ retains only the largest $K$ values of the input vector and sets all remaining entries to zero, consistent with the extraction procedure for the monosemantic activations. For fair comparison, both polysemantic and monosemantic activations are extracted from the same layer and with the same value of $K$.
> * *Monosemantic*: These activations are obtained using the SAE (Sparse Autoencoder) method, which aims to disentangle and sparsify the neuron activations, resulting in more interpretable, monosemantic representations.
> * *w/o SAE*: This refers to the activations before applying the SAE mapping, specifically the hidden states from the penultimate layer of the model.
>
> As shown in Table 1, monosemantic activations produced by the SAE consistently outperform both polysemantic activations and raw hidden states (“w/o SAE”) across all tasks. This demonstrates that disentangling neuron polysemanticity via SAE leads to more effective data selection and superior downstream performance.
>
> ---
>
> ### Trade-off Between Data Quality and Diversity in Target-Specific Selection
> In the context of **target-specific data selection**, our primary goal is to improve model performance on the target task, which is defined by a set of representative examples. To this end, and in line with Less [1], we place greater emphasis on data quality—specifically, by selecting samples that are most similar or relevant to the target examples. While this strategy is effective for optimizing performance on the target task, it inevitably introduces a trade-off: as you have noted, prioritizing data quality can reduce the diversity of the selected data.
>
> To further investigate this trade-off, we explored a strategy to explicitly introduce data diversity into our method, as you suggested. Specifically, whereas our original approach selects samples most similar to the target task based on their SAE representations, our new strategy first ranks candidate samples in descending order of similarity to the target task. We then iterate through this ranked list, maintaining a set of neurons that have already been activated by previously selected samples. A candidate sample is selected only if it activates at least one neuron that has not yet been covered by the current selection. The detailed algorithm is provided in the following pseudocode.
>
> **Algorithm: Diversity-Enhanced Target-Specific Data Selection**
> ```text
> Input:
> - Candidate samples C
> - Target task examples T
> - Similarity function sim(x, T) in SAE representation space
> - Neuron activation function neurons(x)
> Output:
> - Selected sample set S
> 1. For each x in C, compute similarity score s_x = sim(x, T)
> 2. Sort all x in C in descending order by s_x
> 3. Initialize S <- empty, N_covered <- empty
> 4. For each x in sorted C:
>     - Let N_x = neurons(SAE(x))
>     - If N_x is not a subset of N_covered:
>         - Add x to S
>         - Update N_covered <- N_covered union N_x
> 5. Return S
> ```
>
> *Impact of Diversity-Enhancing Strategy on Lexical Diversity of Selected Data* We measure the lexical diversity of the selected data samples using the MTLD (Measure of Textual Lexical Diversity) metric [2], computed with the LexicalRichness [3] package. A higher MTLD score indicates greater lexical diversity. As shown in Table 2, introducing the diversity-enhancing strategy (“MoNA+diversity”) substantially increases the lexical diversity of the selected data across all benchmarks compared to the original target-specific selection (“MoNA”).
>
> **Table 2: Lexical Diversity (MTLD, larger is better) of Selected data**
> | Method                | MMLU  | GSM8K | BBH   | MBPP  | GPQA  |
> | --------------------- | ----- | ----- | ----- | ----- | ----- |
> | RANDOM                | 46.72 | 46.72 | 46.72 | 46.72 | 46.72 |
> | MoNA (ours)           | 54.59 | 28.74 | 40.92 | 27.29 | 32.94 |
> | MoNA+diversity (ours) | 56.83 | 47.58 | 49.13 | 38.61 | 49.13 |
>
> *Impact of Diversity-Enhancing Strategy on Model Performance for the Target Task* Table 3 shows that introducing the diversity-enhancing strategy leads to a decrease in model performance on the target tasks, confirming that increased diversity can come at the cost of reduced accuracy.
>
> **Table 3: Impact of Diversity-Enhancing Strategy on Model Performance**
> | Method                | MMLU  | GSM8K | BBH   | MBPP  | GPQA  | Avg.  |
> | --------------------- | ----- | ----- | ----- | ----- | ----- | ----- |
> | RANDOM                | 64.02 | 58.65 | 63.70 | 46.73 | 30.36 | 52.69 |
> | MONA (ours)           | 64.49 | 67.93 | 66.44 | 48.40 | 31.47 | 55.75 |
> | MoNA+diversity (ours) | 64.78 | 64.71 | 62.40 | 45.20 | 28.35 | 53.09 |
>
> In the next version, we will provide a clearer explanation of the scope of our method within target-specific data selection and offer a more comprehensive discussion of its limitations regarding data diversity.
>
> [1] M. Xia, S. Malladi, S. Gururangan, S. Arora, and D. Chen, “Less: selecting influential data for targeted instruction tuning,” in Proceedings of the 41st International Conference on Machine Learning, ser. ICML’24. JMLR.org, 2025.
>
> [2] McCarthy, Philip M., and Scott Jarvis. "MTLD, vocd-D, and HD-D: A validation study of sophisticated approaches to lexical diversity assessment." Behavior research methods 42.2 (2010): 381-392.
>
> [3] https://github.com/LSYS/LexicalRichness

---

> > ### Comment · Reviewer_Uiye · 2025-07-31
> > **Thank you for answering**
> >
> > My main concerns have been resolved, thus I'm raising my score accordingly.

---

> ### Author Response · Authors · 2025-08-01
>
> Thank you for your positive feedback. We sincerely appreciate your thorough and professional review. Best wishes.

---

### Official Review · Reviewer_TZjR · 2025-06-11

**Clarity:** 4
**Significance:** 3
**Originality:** 4
**Rating:** 4
**Confidence:** 3

**Summary:**

This study introduces a model-centric data selection method for more efficient instruction fine-tuning (fewer data samples) while achieving the same or better performance than existing data selection methods or full fine-tuning.

**Questions:**

1. I would suggest even a preliminary evaluation of the lexical diversity of the selected question, in addition to the quality of the selected data samples.
2. A clearer discussion of the caveats of this method (see weaknesses related to robustness implications) somewhere would be ideal
3. Figure 4 is a bit difficult to interpret -- how many tasks are these patterns representing?

**Ethical Concerns:**

["NO or VERY MINOR ethics concerns only"]

**Final Justification:**

I appreciate the authors for engaging with prior work, but I maintain that generalization and diversity are important for selecting high quality data. I will keep my score.

**Limitations:**

yes

**Quality:**

3

**Strengths And Weaknesses:**

Strengths: (quality, clarity, originality) This work is well written and provides solid experiments and ablations to support the method of data selection based on model activations. Using model activations instead of data-specific features allows more efficient instruction fine-tuning.

Weaknesses: (significance) My main concern with this work is the lack of exploration on the consequences of training with such a method. How does this affect model robustness to unseen instructions (within the same task?).  I see that the quality of the selected data points is evaluated, but there is a missing measurement of diversity here. Prior work shows that instruction-tuned models are sensitive to phrasings seen during training; does this method worsen this problem?

---

> ### Author Rebuttal · Authors · 2025-07-30
>
> ## Response to Reviewer TZjR
>
> Thank you for your insightful and constructive suggestions, which greatly contribute to improving our work. We hope the following point-by-point responses effectively address your concerns.
>
> ---
>
> ### Potential Impact on Robustness and Diversity
>
> In the context of **target-specific data selection**, the primary objective is to enhance model performance on the target task (characterized by a set of representative examples). To achieve this, we following Less [1], naturally place greater emphasis on data quality---specifically, selecting data samples that are most similar or relevant to the representative examples of the target task. This approach aligns with the general principle in machine learning that models tend to generalize better when trained on data closely aligned with the evaluation distribution.
>
> However, we acknowledge that this emphasis on data quality may inadvertently come at the expense of data diversity, as you have pointed out. Following your suggestion, we conducted two additional analyses to more clearly reveal and quantify this effect.
>
> *Robustness to Unseen Instructions* We select GSM8K [2] as the target task and perform instruction tuning using data selected based on GSM8K. We then evaluate the tuned models on both GSM8K (seen instructions) and MATH500 (unseen instructions). While both datasets focus on mathematical reasoning, GSM8K consists of simple, natural-language grade school math problems, whereas MATH500 contains more formal, complex, and diverse problems with distinct instruction styles. This setup enables us to directly test how well each method generalizes to novel instruction phrasings within the same domain. As shown in Table 1, target-specific data selection methods, which emphasize data quality, achieve strong performance on GSM8K but consistently underperform on MATH500, indicating that this focus comes at the cost of generalization to unseen instructions.
>
> **Table 1: Model Performance on GSM8K (Seen) and MATH500 (Unseen)**
> | Method          | GSM8K (seen) | MATH500 (unseen) |
> | --------------- | ------------ | ---------------- |
> | BASE            | 55.50        | 2.40             |
> | FULL            | 65.35        | 21.00            |
> | RANDOM          | 58.65        | 15.33            |
> | MATES           | 54.28        | 19.00            |
> | LESS            | 66.87        | 13.60            |
> | BM25            | 66.64        | 13.80            |
> | DSIR            | 66.94        | 14.20            |
> | DLRDS-BGE       | 64.82        | 16.20            |
> | DLRDS-LLaMA3-8B | 64.75        | 12.40            |
> | MoNA (ours)     | 67.93        | 15.80            |
>
> *Lexical Diversity of Selected Data* We measure the lexical diversity of the selected data samples using the *MTLD* (Measure of Textual Lexical Diversity) metric [4], computed with the LexicalRichness [5] package. A higher MTLD score indicates greater lexical diversity. Table 2 confirm that a stronger emphasis on data quality in target-specific selection indeed leads to reduced data diversity.
>
> **Table 2: Lexical Diversity (MTLD, larger is better) of Selected data**
> | Method          | MMLU  | BBH   | TydiQA |
> | --------------- | ----- | ----- | ------ |
> | RANDOM          | 61.31 | 61.31 | 61.31  |
> | MATES           | 68.25 | 56.83 | 81.01  |
> | LESS            | 76.74 | 53.56 | 77.80  |
> | BM25            | 74.89 | 58.41 | 66.65  |
> | DSIR            | 52.10 | 48.90 | 54.01  |
> | DLRDS-BGE       | 50.79 | 44.63 | 70.63  |
> | DLRDS-LLaMA3-8B | 49.38 | 40.50 | 82.15  |
> | MoNA (ours)     | 66.05 | 42.49 | 81.38  |
>
> In the next version, we will explicitly clarify the scope of our method in target-specific data selection and more thoroughly discuss its limitations in terms of data diversity.
>
> [1] M. Xia, S. Malladi, S. Gururangan, S. Arora, and D. Chen, “Less: selecting influential data for targeted instruction tuning,” in Proceedings of the 41st International Conference on Machine Learning, ser. ICML’24. JMLR.org, 2025.
>
> [2] Cobbe, Karl, et al. "Training verifiers to solve math word problems." arXiv preprint arXiv:2110.14168 (2021).
>
> [3] Lightman, Hunter, et al. "Let's verify step by step." The Twelfth International Conference on Learning Representations. 2023.
>
> [4] McCarthy, Philip M., and Scott Jarvis. "MTLD, vocd-D, and HD-D: A validation study of sophisticated approaches to lexical diversity assessment." Behavior research methods 42.2 (2010): 381-392.
>
> [5] https://github.com/LSYS/LexicalRichness
>
> ---
>
> ### Clarification on Figure 4
> Figure 4 aims to demonstrate that, *in the monosemantic representation space, different tasks tend to activate distinct sets of neurons, while in the polysemantic space, many neurons are activated by multiple tasks.* To this end, we select two tasks---Math and Code---and randomly sample 100 data points from each. For each sample, we visualize the activations across the top-100 most variant neurons. The results show that in the polysemantic space, samples from both tasks frequently produce strong activations on the same neurons (as indicated by the prominent peaks in the upper plot), whereas in the monosemantic space, this overlap is greatly reduced and task-specific activation patterns become much clearer.

---

> ### Author Response · Authors · 2025-08-09
> **Clarification on the Scope of Our Work**
>
> We sincerely thank your thoughtful feedback and valuable insights regarding generalization.
>
> We would like to clarify that **our work follows a line of prior research, including baseline methods such as LESS and DSIR, which aims to improve model performance on a specific target task by selecting high-quality, relevant data. These approaches, including ours, are not designed to select diverse data for enhancing generalization across different tasks.** Therefore, none of these methods are expected to excel at generalizing to novel instructions, and the observed trade-off between data quality and diversity is an inherent limitation of target-specific data selection, rather than a shortcoming unique to our method.
>
> We appreciate the reviewer’s comments and will explicitly clarify this scope and its implications in the revised manuscript.

---

### Official Review · Reviewer_g8cj · 2025-07-02

**Clarity:** 3
**Significance:** 2
**Originality:** 3
**Rating:** 4
**Confidence:** 3

**Summary:**

This paper presents a data selection method for instruction tuning based on a distribution alignment pipeline. The approach selects training examples whose distributions align with the target set. Each sample is represented by its neuronal activation pattern in the model, and sparse autoencoders are used to disentangle polysemantic activations into sparse, monosemantic representations. Finally, the generalized Jaccard similarity is employed to compute embedding similarity. The proposed method (MoNA) outperforms other data selection methods and even surpasses training on the full dataset.

**Questions:**

1. In the experiment shown in Figure 3, how many training examples are sampled under the 5% selection ratio? Whether sampling 10% of the source data can surpass the full data? If not, the practicability of the proposed method is limited, as it overly relies on carefully tuning the sampling ratio for optimal performance.
2. When scaling to 13B models, the superiority of the proposed method diminishes, which raises concerns about its scalability.

**Ethical Concerns:**

["NO or VERY MINOR ethics concerns only"]

**Final Justification:**

I will keep my "borderline accept" rating unchanged. While the rebuttal partially addresses my concerns, I remain concerned about the scalability of the method. On the 13B models, the improvements are not significant, as the proposed method (MONA) surpasses the previous best method (BM25) by less than one point.

**Limitations:**

yes

**Paper Formatting Concerns:**

No formatting concerns.

**Quality:**

3

**Strengths And Weaknesses:**

Strengths
1. The paper is well-written and easy to follow, with clear motivation for each component. The experimental results (as shown in Figure 1(c)) validate the effectiveness of monosemantic activations. Additionally, Figure 6(c) provides valuable insight by demonstrating how the choice of similarity metric can significantly impact performance.
2. Data selection for instruction tuning is a critical topic. The proposed method achieves significant improvements across six benchmarks, outperforming other data selection methods as well as two baseline methods (“full” and “random”).

Weaknesses
1. The method appears sensitive to the data selection ratio, as shown in Figure 3. Sampling 5% of the training data even outperforms sampling 10%, which is somewhat counter-intuitive. In the experiment shown in Figure 3, how many training examples are sampled under the 5% selection ratio?
2. When scaling to 13B models, the superiority of the proposed method diminishes. In Table 2, across five test sets, MoNA achieves the best performance on only two, and lags behind other methods on the remaining three.

---

> ### Author Rebuttal · Authors · 2025-07-30
>
> ## Response to Reviewer g8cj
>
> Thank you very much for your thorough review and valuable feedback. We appreciate the time and effort you have devoted to evaluating our work, as well as your constructive comments and suggestions.
>
> ---
>
> ### Discussion on Performance at Larger Model Scales
>
> We appreciate your observation regarding the diminished superiority of our method when scaling to 13B models, as highlighted in Table 2 of our manuscript. We would like to clarify that **in our original experiments, the SAE used for data selection was trained on LLaMA3-8B, whereas the downstream fine-tuning was performed on LLaMA2-13B**. This introduces a potential mismatch between the SAE model and the target model, which may have affected the observed performance.
>
> To address this concern, we conduct an additional experiment where we train the SAE directly on LLaMA2-13B and used it for data selection. Specifically, following the conclusion from Section 3.5 of our manuscript, we extract neuron activations from the penultimate layer. The SAE is trained using the RedPajama-Data-1T-Sample [1] and the EleutherAI-sparsify training framework [2]. With this adjustment, we observe a further improvement in performance on the 13B model (see Table 1 below, MoNA (LLaMA2-13B SAE) vs. MoNA (LLaMA3-8B SAE)), demonstrating that using a matched SAE and target model is important for maintaining the effectiveness of our approach at larger scales.
>
> **Tabel 1:Performance Comparison of Data Selection Methods on LLaMA2-13B**
> | Method                      | MMLU  | GSM8K | BBH   | MBPP  | GPQA  | Avg.  |
> | --------------------------- | ----- | ----- | ----- | ----- | ----- | ----- |
> | BASE                        | 55.11 | 24.03 | 46.74 | 27.00 | 30.58 | 36.69 |
> | FULL                        | 57.61 | 55.95 | 52.63 | 35.00 | 27.01 | 45.64 |
> | RANDOM                      | 55.96 | 41.50 | 51.37 | 31.13 | 28.64 | 41.72 |
> | MATES                       | 55.68 | 37.07 | 51.17 | 31.20 | 26.34 | 40.29 |
> | LESS                        | 60.38 | 48.75 | 50.42 | 25.80 | 27.90 | 42.65 |
> | BM25                        | 57.60 | 58.15 | 52.65 | 34.60 | 27.90 | 46.18 |
> | DSIR                        | 55.83 | 53.53 | 52.02 | 31.60 | 27.23 | 44.04 |
> | DLRDS-BGE                   | 56.65 | 56.63 | 52.34 | 35.60 | 27.68 | 45.78 |
> | DLRDS-LLaMA3-8B             | 58.65 | 52.31 | 52.36 | 35.20 | 26.56 | 45.02 |
> | LLM2Vec                     | 57.02 | 58.30 | 51.27 | 34.80 | 27.90 | 45.86 |
> | MONA (ours, LLaMA3-8B SAE)  | 57.26 | 60.27 | 52.23 | 35.60 | 27.90 | 46.65 |
> | MONA (ours, LLaMA2-13B SAE) | 58.73 | 60.12 | 52.48 | 36.60 | 27.90 | 47.17 |
>
>
>
> [1] https://huggingface.co/datasets/togethercomputer/RedPajama-Data-1T-Sample
>
> [2] https://github.com/EleutherAI/sparsify/tree/main
>
> ---
>
> ### On the Sensitivity of Data Selection Ratio
> * *Experimental Details of Figure 3 in Our Manuscript* In our exploration of the data selection ratio, we used the LESS dataset [1], which contains a total of 270,699 examples. Specifically,
>     * A 5% selection ratio corresponds to 13,534 sampled examples.
>     * On this dataset, training the model with 10% of the data does not outperform using 5%; moreover, the 10% subset also does not surpass the performance achieved by using the full dataset. This trend holds true for all baseline methods evaluated, as well as for our proposed approach.
> * *Additional Validation on OpenHermes2.5* We believe that the counter-intuitive phenomenon---where using 10% of the data results in worse performance than using 5%---is related to the characteristics of the dataset. To further investigate this, we conduct the same experiments on the OpenHermes2.5 dataset (1001551 samples) [2]. On OpenHermes2.5, we observe that using 10% of the data leads to better performance than using 5% (see Table 2 bellow). This indicates that the phenomenon observed on LESS is data-dependent and does not necessarily occur on other datasets.
>
> **Table 2: Performance of MoNA with Different Data Selection Ratios on OpenHermes2.5 (Using LLaMA3.1-8B)**
> | Method      | Ratio | MMLU  | GSM8K | BBH   | MBPP  | GPQA  | Avg.  |
> | ----------- | ----- | ----- | ----- | ----- | ----- | ----- | ----- |
> | BASE        | -     | 65.30 | 55.50 | 63.08 | 46.40 | 28.12 | 51.68 |
> | FULL        | 100%  | 64.60 | 65.35 | 64.31 | 49.00 | 27.90 | 54.23 |
> | MoNA (ours) | 5%    | 64.49 | 67.93 | 66.44 | 48.40 | 31.47 | 55.75 |
> | MoNA (ours) | 10%   | 64.58 | 68.16 | 66.63 | 49.60 | 31.03 | 56.00 |
>
> [1] M. Xia, S. Malladi, S. Gururangan, S. Arora, and D. Chen, “Less: selecting influential data for targeted instruction tuning,” in Proceedings of the 41st International Conference on Machine Learning, ser. ICML’24. JMLR.org, 2025.
>
> [2] Teknium, “Openhermes 2.5: An open dataset of synthetic data for generalist llm assistants,” 2023. [Online]. Available: https://huggingface.co/datasets/teknium/OpenHermes-2.5

---

> > ### Comment · Reviewer_g8cj · 2025-08-05
> >
> > Thank you for your responses. The results presented in Table 1 partially address my concerns. I also appreciate the additional results in Table 2, and agree that the optimal data selection ratio is indeed data-dependent. However, on the13B models, the improvement over the previous methods is relatively small; for example, MONA surpasses the previous best method (BM25) by less than one point. Therefore, I will keep my scores unchanged.

---

> ### Author Response · Authors · 2025-08-05
>
> Thank you very much for your recognition of the parts of our rebuttal and for raising this interesting point. We appreciate your careful review and thoughtful feedback.
>
> We have noticed, as you pointed out, that the improvement of our method over the strongest baseline (BM25) is relatively modest on the 13B models. Upon further analysis, we found that our method achieves an average improvement of **1.7** points over BM25 on the **MMLU, GSM8K, and MBPP** tasks. However, for the BBH and GPQA benchmarks, all data selection methods (including ours) perform similarly, and the differences are minimal. As a result, when averaging across all five tasks, the overall improvement is diluted to 0.99 points.
>
> We suspect that the similar performance among all methods on BBH and GPQA may be related to the model itself. Supporting this, on the LLaMA3.1-8B model, our method (MONA) achieves an average improvement of **2.39** points over BM25 on these two tasks (see Table 1 in our manuscript).
>
> Thank you again for your valuable comments, which have prompted us to reflect more deeply on the interplay between data selection and model performance.

---

### Official Review · Reviewer_wUy5 · 2025-07-03

**Clarity:** 4
**Significance:** 3
**Originality:** 3
**Rating:** 5
**Confidence:** 5

**Summary:**

The paper proposes a new method of data selection for task-specific fine-tuning by leveraging sparse autoencoders to obtain monosemantic activations for selecting task-specific training data. The method selects training data by comparing the similarity between the monosemantic embedding of a given data point and that of the task prototype data. Extensive experiments on task-specific fine-tuning show that the proposed method outperforms multiple data selection baselines from the literature.

**Questions:**

See the weakness above.

**Ethical Concerns:**

["NO or VERY MINOR ethics concerns only"]

**Final Justification:**

The new experiments the authors added in the rebuttal response addressed both of my concerns.

**Limitations:**

Yes

**Quality:**

4

**Strengths And Weaknesses:**

Strengths

1. The paper proposes a novel data selection method with a good motivation for disentangling polysemantic embeddings of training data examples into monosemantic embeddings.

2. The paper designs a general distribution alignment pipeline and explores different critical design choices within the pipeline.

3. The paper conducts thorough experiments, compares with a wide range of well-established data selection baselines from the literature, and shows strong empirical results across the board.

4. The paper provides thorough ablation analyses of the method, potentially leading to future work in this direction.

Weakness

1. There is a lack of analyses on the impact of the SAE model. The main novelty of the paper lies in using SAE to embed the training data, but it is unclear in the experiments if the quality of the SAE embeddings, especially how monosemantic/sparse the SAE embeddings are, has an impact on the quality of data selection. However, the paper only uses the same one SAE model for all experiments.

2. There is a lack of comparisons with other monosemantic embeddings. This is not a critical weakness, but I can imagine applying the same workflow but swapping SAE with another model that can generate relatively monosemantic embeddings, especially MoE models: prior work has shown that MoE models can be used effectively as text embedders out of the box [1] and experts of pretrained MoE can be specialized to process a very specific set of tokens [2], leading to potentailly monosemantic representations.

[1] Ziyue Li, Tianyi Zhou. Your Mixture-of-Experts LLM Is Secretly an Embedding Model For Free. https://arxiv.org/abs/2410.10814

[2] Niklas Muennighoff, Luca Soldaini, Dirk Groeneveld, Kyle Lo, Jacob Morrison, Sewon Min, Weijia Shi, Pete Walsh, Oyvind Tafjord, Nathan Lambert, Yuling Gu, Shane Arora, Akshita Bhagia, Dustin Schwenk, David Wadden, Alexander Wettig, Binyuan Hui, Tim Dettmers, Douwe Kiela, Ali Farhadi, Noah A. Smith, Pang Wei Koh, Amanpreet Singh, Hannaneh Hajishirzi. OLMoE: Open Mixture-of-Experts Language Models. https://arxiv.org/abs/2409.02060

---

> ### Author Rebuttal · Authors · 2025-07-30
>
> ## Response to Reviewer wUy5
> Thank you very much for your insightful and constructive comments, which have greatly contributed to improving the quality of our work.
>
> ---
>
> ### Supplementary Analysis Using an Additional SAE Model
> Following your suggestion, we train a new SAE model on a larger backbone (LLaMA2-13B). Specifically, we follow the conclusion from Section 3.5 of our manuscript, extracting neuron activations from the penultimate layer. The training data used is *RedPajama-Data-1T-Sample* [1], and we utilize the *EleutherAI-sparsify* [2] training framework for SAE training.
>
> **Table 1: Results of Using Different SAE Models on LLaMA2-13B**
> | Method     | SAE Model   | MMLU  | GSM8K | BBH   | MBPP  | GPQA  | Avg.  |
> | ---------- | ----------- | ----- | ----- | ----- | ----- | ----- | ----- |
> | BASE       | -           | 55.11 | 24.03 | 46.74 | 27.00 | 30.58 | 36.69 |
> | FULL       | -           | 57.61 | 55.95 | 52.63 | 35.00 | 27.01 | 45.64 |
> | MoNA(ours) | LLaMA-3-8B  | 57.26 | 60.27 | 52.23 | 35.60 | 27.90 | 46.65 |
> | MoNA(ours) | LLaMA-2-13B | 58.73 | 60.12 | 52.48 | 36.60 | 27.90 | 47.17 |
>
> As shown in Table 1, we compare the performance of our method (MoNA) using different SAE models (trained over LLaMA-3-8B vs. LLaMA-2-13B), alongside the BASE and FULL baselines. The results demonstrate a larger SAE backbone can further enhance the effectiveness of data selection and downstream performance. These findings suggest that the quality and scale of the SAE model have a positive impact on the overall results.
>
> [1] https://huggingface.co/datasets/togethercomputer/RedPajama-Data-1T-Sample
>
> [2] https://github.com/EleutherAI/sparsify/tree/main
>
> ---
>
> ### Comparison with MoE-based Monosemantic Embeddings
>
> We greatly appreciate this insightful and stimulating suggestion. Following your advice, we replace SAE with MoE-based embeddings, specifically adopting the approach described in your referenced work [1]: for each token, we use the concatenation of expert routing scores from all layers of a MoE model (obtained during a forward pass) as the token’s embedding. For this experiment, we use the MoE model you recommended, *allenai/OLMoE-1B-7B-0924* [2]. Apart from substituting the SAE-based embeddings with MoE-based embeddings, all other components and settings of our workflow were kept the same to ensure a fair comparison.
>
> **Table 2: Evaluation of Monosemantic Embeddings: SAE-based vs. MoE-based**
> | Method     | Embedding | MMLU  | GSM8K | BBH   | MBPP  | GPQA  | Avg.  |
> | ---------- | --------- | ----- | ----- | ----- | ----- | ----- | ----- |
> | BASE       | -         | 65.30 | 55.50 | 63.08 | 46.40 | 28.12 | 51.68 |
> | FULL       | -         | 64.60 | 65.35 | 64.31 | 49.00 | 27.90 | 54.23 |
> | MoNA(ours) | SAE-based | 64.49 | 67.93 | 66.44 | 48.40 | 31.47 | 55.75 |
> | MoNA(ours) | MoE-based | 64.02 | 65.43 | 63.83 | 49.00 | 31.03 | 54.66 |
>
> As shown in Table 2, under our current experimental setup, using MoE-based embeddings yields a **modest** improvement over the FULL baseline (54.66 vs. 54.23 in average score), indicating that monosemantic representations from MoE models can provide a slight benefit for task-specific data selection. However, MoE-based embeddings still lag behind SAE-based embeddings by a noticeable margin, as reflected in the average scores (55.75 vs. 54.66).
>
> Based on our experience, we suspect several factors may contribute to this difference:
> * *MoE-based embedding utilization in data selection requires further investigation.* The effectiveness of MoE-based embeddings for task-specific data selection may depend on design choices such as which layer’s expert routing scores to use and how to measure similarity in the MoE embedding space. Our current approach simply concatenates routing scores from all layers, but more targeted strategies (e.g., selecting specific layers or aggregating expert activations differently) may yield better results.
> * *MoE-based embeddings may primarily capture domain-level monosemanticity [2] rather than fine-grained semantic distinctions.* For example, arXiv is a common source of training data for language models and contains a wide variety of text, including both academic citations and HTTP request patterns. While SAE-based embeddings are able to distinguish between the semantics of these different types of content [3], MoE models may route all tokens from arXiv data to the same expert [2], regardless of such finer-grained differences. This suggests that the monosemanticity provided by MoE-based embeddings is more aligned with broad domain or topic separation, and may not capture the more nuanced distinctions needed for our data selection workflow.
>
> Certainly, these are preliminary hypotheses informed by our previous experience and the rapid experiments we conducted based on your valuable suggestion. More comprehensive experiments and analysis are required. We believe that with further exploration and optimization, the potential of MoE-based embeddings for data selection can be better realized, and we consider this a promising direction for future work.
>
> [1] Ziyue Li, Tianyi Zhou. Your Mixture-of-Experts LLM Is Secretly an Embedding Model For Free. https://arxiv.org/abs/2410.10814
>
> [2] Niklas Muennighoff, Luca Soldaini, Dirk Groeneveld, Kyle Lo, Jacob Morrison, Sewon Min, Weijia Shi, Pete Walsh, Oyvind Tafjord, Nathan Lambert, Yuling Gu, Shane Arora, Akshita Bhagia, Dustin Schwenk, David Wadden, Alexander Wettig, Binyuan Hui, Tim Dettmers, Douwe Kiela, Ali Farhadi, Noah A. Smith, Pang Wei Koh, Amanpreet Singh, Hannaneh Hajishirzi. OLMoE: Open Mixture-of-Experts Language Models. https://arxiv.org/abs/2409.02060
>
> [3] T. Bricken, A. Templeton, J. Batson, B. Chen, A. Jermyn, T. Conerly, N. Turner, C. Anil, C. Denison, A. Askell, R. Lasenby, Y. Wu, S. Kravec, N. Schiefer, T. Maxwell, N. Joseph, Z. Hatfield-Dodds, A. Tamkin, K. Nguyen, B. McLean, J. E. Burke, T. Hume, S. Carter, T. Henighan, and C. Olah, “Towards monose367 manticity: Decomposing language models with dictionary learning,” Transformer Circuits Thread, 2023, https://transformer-circuits.pub/2023/monosemantic-features/index.html.

---

> > ### Comment · Reviewer_wUy5 · 2025-08-05
> >
> > I would like to thank the reviewers for the detailed rebuttal and including additional experiments. They addressed my concerns, and I would like to increase my confidence to 5 in my already positive score of 5. Thanks for the great work.

---

> ### Author Response · Authors · 2025-08-06
>
> Thank you very much for your reply and recognition. Your insightful suggestions have offered us profound guidance and inspiration for our work. We truly appreciate your dedication as a reviewer. Best regards!

---

### Note · Authors · 2025-08-12

We sincerely thank the reviewers, area chairs, senior area chairs, and program chairs for their time and insightful feedback during the review process.

**All reviewers initially gave positive evaluations (5, 4, 4, and 4) and maintained positive attitudes ($\ge4$) after we addressed their concerns with detailed experiments and clarifications**. In particular, one reviewer raised the score (from an initial 4), another increased the confidence to the highest level (5) while maintaining a positive score (5), and the remaining reviewers kept their positive scores (4).

The initial reviews highlighted the following strengths and weaknesses:

#### Strengths
* *Motivation*: clear motivation for disentangling polysemantic embeddings in data selection (wUy5, g8cj)
* *Method*: novel, simple, and intuitive monosemantic embedding-based approach (wUy5, g8cj, Uiye)
* *Experiments*: comprehensive experiments and ablations with strong results (ALL)
* *Presentation*: well-written and easy to follow (g8cj, TZjR, Uiye)

#### Weaknesses
* Limited analysis of SAE's impact and comparison with other monosemantic embeddings (wUy5)
* Sensitivity to data selection ratio, with counter-intuitive results and modest performance gains on larger models (g8cj)
* Limited exploration of robustness to unseen instructions and insufficient discussion of data diversity (TZjR)
* Need for clearer discussion of quality-diversity trade-off and comparison between SAE and raw activations (Uiye)

After rebuttal, reviewer **Uiye** raised the score and reviewer **wUy5** increased the confidence after confirming all issues were addressed. Reviewer **g8cj** kept the positive score, noting modest improvements on larger models; we clarified that similar performance across all methods on two benchmarks diluted the overall improvement. Reviewer **TZjR** also maintained the positive score after reviewing our additional experiments on generalization and data diversity, which showed that all methods performed similarly and faced challenges in these areas. We clarified that, consistent with prior work such as LESS and DSIR, our method is designed for target-specific data selection, and these limitations are inherent to this research direction. Enhancing generalization across diverse tasks is outside the main scope of our study.

We appreciate the feedback, which helped us refine our work and clarify its scope. We will incorporate these improvements into the next version.

---

### Decision · Program_Chairs · 2025-09-17

**Decision:**

Accept (poster)

**Comment:**

The paper introduces a data selection method, MoNA, for efficient instruction tuning. It represents data samples using internal model activations, which are disentangled into sparse, monosemantic representations via a sparse autoencoder (SAE). This method is shown to consistently outperform existing data selection baselines.

Overall, the paper is viewed as a solid, well-supported contribution to the field of data selection for instruction tuning.